# Chromosome instability induced by a single defined sister chromatid fusion

Katsushi Kagaya[1,2], Naoto Noma-Takayasu[3,*] , Io Yamamoto[3,*], Sanki Tashiro[3], Fuyuki Ishikawa[3] , Makoto T Hayashi[1,3]

Chromosome fusion is a frequent intermediate in oncogenic chromosome rearrangements and has been proposed to cause multiple tumor-driving abnormalities. In conventional experimental systems, however, these abnormalities were often induced by randomly induced chromosome fusions involving multiple different chromosomes. It was therefore not well understood whether a single defined type of chromosome fusion, which is reminiscent of a sporadic fusion in tumor cells, has the potential to cause chromosome instabilities. Here, we developed a human cell-based sister chromatid fusion visualization system (FuVis), in which a single defined sister chromatid fusion is induced by CRISPR/Cas9 concomitantly with mCitrine expression. The fused chromosome subsequently developed extra-acentric chromosomes, including chromosome scattering, indicative of chromothripsis. Live-cell imaging and statistical modeling indicated that sister chromatid fusion generated micronuclei (MN) in the first few cell cycles and that cells with MN tend to display cell cycle abnormalities. The powerful FuVis system thus demonstrates that even a single sporadic sister chromatid fusion can induce chromosome instability and destabilize the cell cycle through MN formation.

## Introduction

Chromosome abnormalities are at the core of tumorigenesis. Among oncogenic chromosomal rearrangements, chromosome fusion that gives rise to a dicentric chromosome is highly deleterious because of the generation of unresolved chromatin bridges after anaphase (Maciejowski & de Lange, 2017). Previous studies led to the hypotheses that chromosome fusions cause multiple tumor-driving abnormalities, including breakage-fusion-bridge cycle (Ishikawa, 1997; Maser & Depinho, 2002), binucleation (Pampalona et al, 2012), chromothripsis and kataegis (Maciejowski et al, 2015), mitotic arrest (Hayashi et al, 2015), and cGAS/STING activation (Nassour et al, 2019). In these studies, the effects of chromosome fusions have been analyzed by artificial disruption of telomere-binding proteins that protect the chromosome ends from activating DNA damage response. Among the telomere-binding complex called shelterin, TRF2 is central in telomere protection and targeted by various methods including dominant-negative allele (van Steensel et al, 1998), shRNA-dependent knockdown (Takai et al, 2003; Cesare et al, 2013), and cre-loxP- and CRISPR/Cas9-mediated knockout (Celli & de Lange, 2005; Hayashi et al, 2015) to understand the consequence of chromosome fusions. The fate of chromosome fusion has also been analyzed during telomere crisis induced by replicative telomere shortening in p53-compromised cells and mice that lack functional telomerase (Shay et al, 1991; Counter et al, 1992; Blasco et al, 1997; Chin et al, 1999). Disruption of TRF2 results in massive chromosome fusion events (Celli & de Lange, 2005; Hayashi et al, 2015), whereas ongoing telomere shortening gives rise to the continuous emergence of dicentric chromosomes (Counter et al, 1992). Thus, in both experimental systems, multiple chromosome fusions are induced over time to a different extent, which makes it challenging to analyze the effect of a single chromosome fusion event. Besides, there are at least three different types of end-to-end chromosome fusion induced in these systems. Inter-chromosomal fusion involves chromosome ends of two distinct chromosomes, whereas intra-chromosomal fusion occurs between both ends of the same chromosome, resulting in a ring-shaped chromosome. The third is sister chromatid fusion (SCF) that requires each end of sister chromatid pair after DNA replication. Among these, SCF has been implicated in the escape from telomere crisis by inducing the appropriate genetic alterations (Jones et al, 2014), suggesting that

[1]The Hakubi Center for Advanced Research, Kyoto University, Yoshida-Konoe-cho, Kyoto, Japan   [2]Seto Marine Biological Laboratory, Field Science, Education and Research Center, Kyoto University, Wakayama, Japan   [3]Department of Gene Mechanisms, Graduate School of Biostudies, Kyoto University, Yoshida-Konoe-cho, Kyoto, Japan

Correspondence: hayashi.makoto.8a@kyoto-u.jp
*Naoto Noma-Takayasu and Io Yamamoto contributed equally to this work
Katsushi Kagaya's present address is Center for Education and Research in Information Science and Technology (CERIST), Graduate School of Information Science and Technology, The University of Tokyo, Tokyo, Japan
Sanki Tashiro's present address is Institute of Molecular Biology, University of Oregon, Eugene, OR, USA
Makoto T Hayashi's present address is IFOM-KU Joint Research Laboratory, Department of State-of-the-art and International Medicines, Graduate School of Medicine, Kyoto University, Yoshida-Konoe-cho, Kyoto, Japan

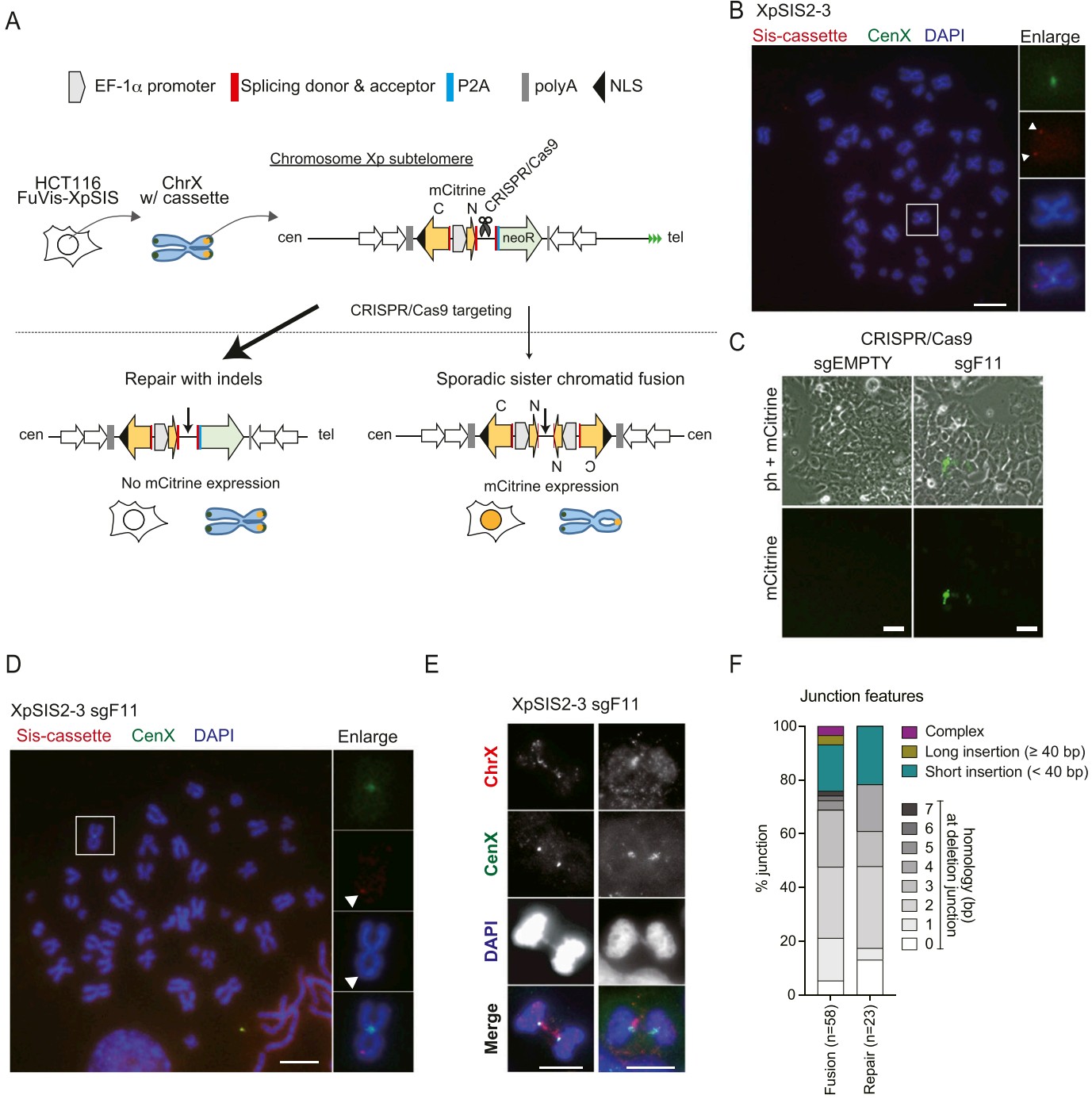

**Figure 1.   Validation of CRISPR/Cas9-mediated sister chromatid fusion (SCF) visualization system.**
**(A)** Schematic overview of the development of the FuVis-XpSIS system. Targeting the spacer region between the N-terminus of *mCitrine* and the *neoR* gene by CRISPR/Cas9 results in either repair with indels (left) or sporadic SCF and full-length mCitrine expression (right). **(B)** FISH image of mitotic chromosomes of XpSIS2-3 cells showing the sister cassette (red), the X centromere (green), and DAPI (blue). Colors were adjusted on individual and merged images. Arrowheads indicate sister cassette signals. Scale bar: 10 μm. **(C)** Phase-contrast and fluorescence images of XpSIS2-3 cells 10 d postinfection with lentivirus carrying CRISPR/Cas9 and indicated sgRNA. Scale bar: 50 μm. **(B, D)** FISH image of mitotic chromosomes of XpSIS2-3 sgF11 cells, shown as in (B). mCitrine-positive cells were sorted 10 d postinfection with lentivirus carrying CRISPR/Cas9 and sgF11. Arrowhead indicates the sister cassette signal (red) and SCF (DAPI). Scale bar: 10 μm. **(E)** FISH images of XpSIS2-3 sgF11 cells at 10 d postinfection. Chromosome bridges of the X chromosome were visualized by the whole X chromosome (red) and the X centromere (green) probes and DAPI-stained chromosomes (blue). Colors were adjusted on individual and merged images. Scale bar: 10 μm. **(F)** Bar graph represents percentage of indicated junction features for SCF and repair with indels in mCitrine-positive and whole population of XpSIS2-3 sgF11 cells, respectively, at 10 d postinfection.
Source data are available for this figure.

each type of chromosome fusion has a distinct effect on cellular fitness. However, it was complicated to analyze the types and the number of fusions in a given cell without harvesting the cell, and the exact timing of the fusion events was also exceedingly difficult to discern. A recently developed technique that uses sequence-specific nucleases such as I-SceI and TALEN to induce double-strand break (DSB) in the subtelomere region can potentially regulate the number of fusion as a consequence of abnormal repair between two distinct subtelomeric DSB (Lo et al, 2002; Liddiard et al, 2016). However, the nuclease-mediated method still failed to control the types of fusion and the timing of its induction.

Here, we have developed a cell-based chromosome fusion visualization (FuVis) system, which visualizes an SCF at the Xp subtelomere (FuVis-XpSIS). The FuVis-XpSIS relies on an artificial cassette integrated into the Xp subtelomere. The cassette has been designed so that the CRISPR/Cas9-mediated DSB of the cassette generates a single SCF concomitantly with mCitrine expression. The cytological analysis suggested that an SCF causes extra-acentric X chromosome abnormalities. Live-cell imaging and a lineage tracking of the mCitrine-positive cells suggested that a single SCF causes MN formation. Our statistical modeling approach indicates that a single SCF increases the probability of MN formation by 10.3 times. The analysis also indicates that MN delay the average duration of interphase of the cell cycle by 2.3 times, and that MN-positive cells possess more abnormalities than their MN-negative sister lineages. These results illuminate that the FuVis is a potent tool to follow the fate of a single defined DNA rearrangement in living cells. We propose that even a single sporadic SCF in tumor cells potentially causes a deleterious effect on cellular fitness through MN formation.

## Results

### Development of fusion visualization system

To overcome the limitations of the conventional methods, we designed a system named Fusion Visualization system for Xp SCF (FuVis-XpSIS) (Fig 1A). The FuVis-XpSIS uses a DNA cassette that harbors an *mCitrine* gene that is interrupted by a splicing donor and acceptor (Fig 1A). The N terminus is driven by an EF-1 promoter, whereas the C terminus is located upstream of the EF-1a promoter in the opposite orientation. The region downstream of the N terminus harbors a spacer region that contains multiple useable CRISPR/Cas9 target sequences that are absent from the human genome (Table S1), and a splicing acceptor– and self-cleaving peptide sequence (P2A)-tagged neomycin resistance (*neoR*) gene. The entire cassette sequence was flanked by tandem cHS4 insulators to suppress spreading of silent chromatin into the cassette (West et al, 2004), and integrated into a telomere-adjoining subtelomeric locus on the short arm of the X chromosome in male-derived HCT116 cells by homology-mediated recombination (HR) via CRISPR/Cas9 targeting (Fig 1A). We reasoned that targeting of the spacer region between the N terminus of mCitrine and the *neoR* gene by CRISPR/Cas9 would result in either indel by erroneous repair at the target locus or a sporadic SCF, concomitant with an expression of the full-length *mCitrine* gene (Fig 1A). Through two independent HR-mediated integrations of the cassette, we isolated 24 and 48 G418-positive clones, respectively. We

validated the integration by genomic PCR and obtained 11 candidate clones that showed the expected size of the PCR product (FuVis-XpSIS1-15, 1-21, 2-3, 2-6, 2-9, 2-13, 2-21, 2-33, 2-36, 2-38, and 2-39) (Fig S1A and B and Table S2). We performed quantitative PCR (qPCR) to analyze copy numbers of the integrated cassette in these clones. For this purpose, an AAVS1 sequence on chromosome 19 was cloned into a plasmid carrying the sister cassette, which was used as a qPCR standard template (Fig S1C). We found that most clones harbor multiple copies of the sister cassette compared with the AAVS1 sequence and that (FuVis-)XpSIS2-3 clone carries a single copy of the cassette (Fig S1D). The qPCR and dual-colored FISH analysis using DNA probes spanning the entire X chromosome and the X centromere confirmed that the XpSIS2-3 clone harbors a single X chromosome (Fig S1E and F). The integration of the cassette into the Xp subtelomere was further confirmed by Southern hybridization (Fig S1G and H) and FISH using a DNA probe specific to the cassette (Fig 1B). We realized that about 30% of the XpSIS2-3 cells show translocation at the sister cassette site of the X chromosome (Fig S1I and J). FISH revealed the sister cassette remains on the translocated X chromosome (Fig S1K). Because such translocation at the telomere-proximal side of the cassette does not affect the mechanism of the system (Fig 1A), we proceeded to further validation of the XpSIS2-3. We evaluated the efficiency of mCitrine expression by targeting different spacer sequences (sgFUSIONs) and found that the most efficient inducer of mCitrine was sgFUSION11 (hereafter sgF11) (Figs 1C and S2A and Table S1), which we chose for the subsequent analyses. The induction of an SCF was confirmed by FISH on mitotic spreads of mCitrine-positive XpSIS2-3 cells expressing CRISPR/Cas9-sgF11 (XpSIS2-3 sgF11) after sorting (Figs 1D and S2B). Consistent with the induction of an SCF, the dual-colored FISH on interphase XpSIS2-3 sgF11 cells revealed DNA bridge formations on X chromosomes between two neighboring cells (Fig 1E).

### SCF junction analysis

To further validate the formation of an SCF, fusion junctions and repair junctions (Fig S2C) were amplified using genomic DNA extracted from mCitrine-positive and the entire population of XpSIS2-3 sgF11 cells, respectively (Fig S2D). Alignment of cloned sequences to expected fusion and repair junctions in the absence of any indels indicated that SCF, but not repair junction, coincided with large deletions (Fig S2E). The SCF junctions preferentially involved longer microhomology than the repair products (Fig 1F). The preference of large deletions, microhomologies, and short insertion at fusion junctions suggests that SCFs are processed by microhomology-mediated end joining (MMEJ) (Sfeir & Symington, 2015), which is active in the late S/G2 phase when sister chromatids are present (Yang et al, 2018). This profile of fusion junctions is consistent with naturally occurring and TALEN-induced chromosome end-to-end fusions (Capper et al, 2007; Letsolo et al, 2010; Tankimanova et al, 2012; Liddiard et al, 2016).

### Construction of the control system

We also designed a control system, in which a control cassette sequence was integrated into the Xp subtelomeric locus (Fig 2A). The control cassette contains the P2A and the splicing acceptor–tagged

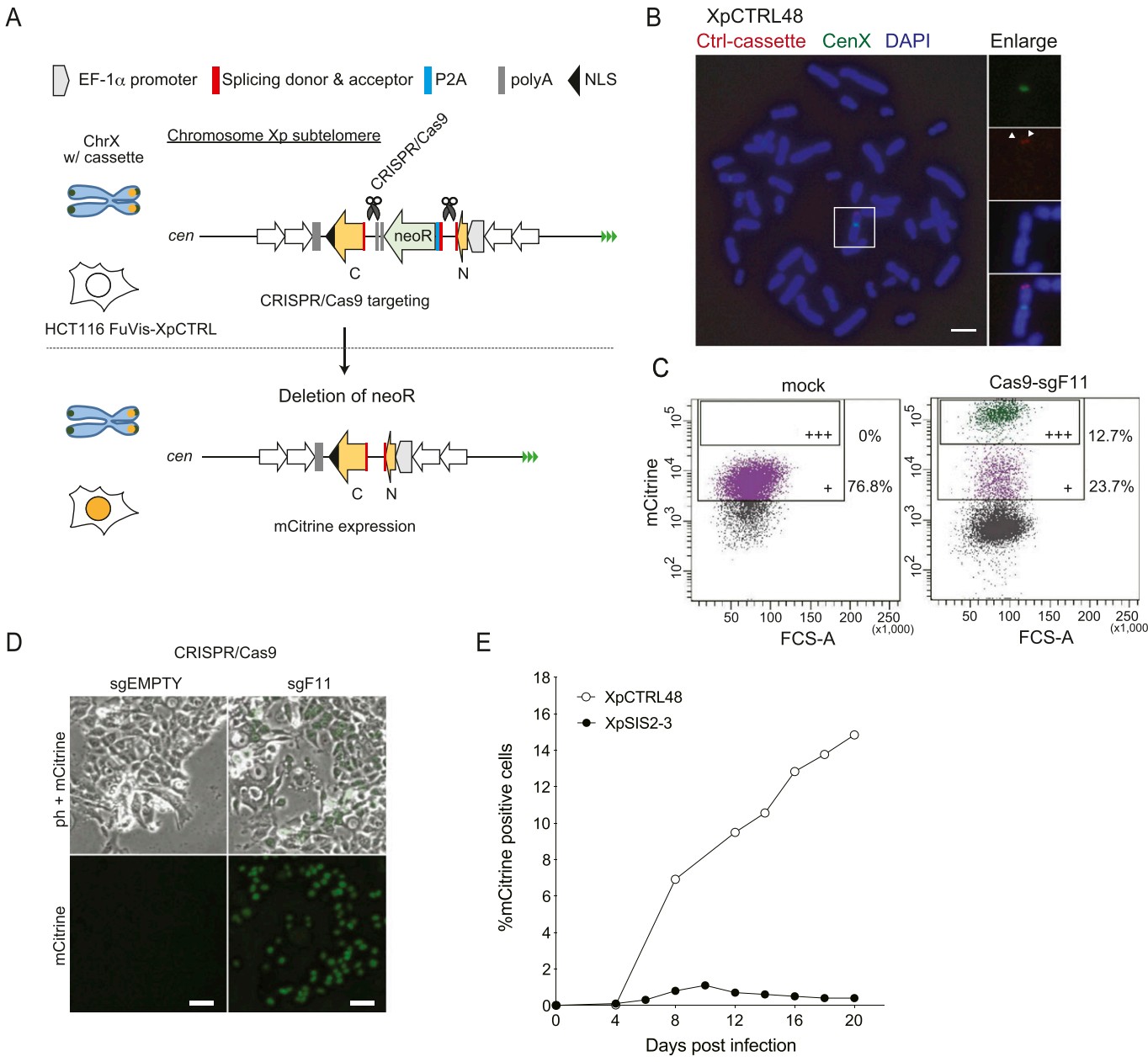

**Figure 2. Validation of FuVis-Xp control system.**
**(A)** Schematic overview of the FuVis-XpCTRL system. CRISPR/Cas9-mediated removal of the *neoR* results in mCitrine expression. **(B)** FISH images of mitotic chromosomes of XpCTRL48 cells, shown as in Fig 1B. Scale bar: 10 *μm*. **(C)** Flow cytometry analysis of XpCTRL48 cells expressing CRISPR/Cas9 and sgF11 at 10 d post lentivirus infection. +: background level of mCitrine expression. +++: mCitrine expression induced by CRISPR/Cas9 and sgF11. **(D)** Phase-contrast and mCitrine fluorescence images of XpCTRL48 cells at 10 d postinfection with lentivirus carrying CRISPR/Cas9 and indicated sgRNA. Scale bar: 50 *μm*. **(E)** Time course of mCitrine expression in indicated cells expressing Cas9 and sgF11. The results were reproducible in two independent experiments.

*neoR* gene and two tandem polyA sequences franked by the N terminus and the C terminus of *mCitrine* in the same orientation. CRISPR/Cas9 targeting of the upstream and downstream of the *neoR* gene results in a sporadic deletion of the *neoR* gene and mCitrine expression (Fig 2A). Among 48 G418-resistant clones obtained upon the integration of the control cassette, three clones (named as FuVis-XpCTRL16, 33, and 48) were positive for PCR products indicating targeted integration (Fig S3A and B). The copy number analysis by qPCR revealed that the XpCTRL48 clone harbors a single copy of the cassette (Fig S3C). Southern hybridization and sequencing of genomic PCR products revealed that XpCTRL48 possessed a duplication of a telomere-distal homology arm sequence (Fig S3D and E and data not shown). Because the duplication of the homology arm does not affect the mechanism of the system, we chose XpCTRL48 cells for subsequent analysis. The integration was further confirmed by FISH (Figs 2B and S3F).

XpCTRL48 cells possessed a background level of mCitrine expression (Fig 2C). Introduction of CRISPR/Cas9-sgF11 into XpCTRL48, however, induced robust expression of mCitrine (Fig 2C and D).

We performed sequencing analysis of repair junctions using genomic DNA from the whole population of XpCTRL48 sgF11 cells (Fig S3G). The PCR amplification generated a large product that corresponds to the original sequence and faint short products corresponding to repaired products with deletion (Fig S3H); the latter was cloned for sequencing analysis. The sequenced clones possessed two types of junction: those that completely lost sequences between the CRISPR/Cas9 targets (type I) and those that partially lost the sequences (type II) (Fig S3I). Both types of junction possessed a signature of MMEJ, which was indicated by extensive truncations, microhomologies, and insertions at the junction (Fig S3J and K) (Sfeir & Symington, 2015), albeit longer insertion in type I implies different mechanism. These results suggest that both XpSIS and XpCTRL mostly rely on MMEJ for the targeted genome rearrangements required for mCitrine expression.

### Kinetics of mCitrine expression upon CRISPR/Cas9 targeting

To analyze the kinetics of mCitrine expression following SCF, XpCTRL48 sgF11, and XpSIS2-3 sgF11 cells were cultured for 3 wk postinfection. Flow cytometry analysis revealed that the percentage of mCitrine-positive cells gradually increased in XpCTRL48 over a long-term culture period, whereas it peaked at 10 d postinfection in XpSIS2-3, and gradually decreased until it reached a plateau around 14 d postinfection (Fig 2E). This kinetics is consistent with an assumption that, upon SCF, a single *mCitrine* gene is generated in G2 phase, which can be propagated to either one of two daughter cells following the first mitosis, whereas the *neoR* deletion in XpCTRL48 cells resulted in two *mCitrine* genes in G2 phase (Fig S4A and B). To address this assumption, mCitrine-positive XpCTRL48 and XpSIS2-3 sgF11 cells at day 8 postinfection were sorted, cultured for another 25 d and analyzed by FACS (Fig S4C). Strikingly, 88.7% of XpCTRL48 sgF11 cells were still mCitrine positive, whereas only 41.3% of XpSIS2-3 sgF11 cells were mCitrine positive (Fig S4D and E), indicating that the *mCitrine* gene was not propagated to both daughter cells in XpSIS2-3 sgF11 cells. In both XpCTRL48 and XpSIS2-3, about 10% of cells became mCitrine-negative additionally after re-culturing (assuming 100% and 50% are expected values for XpCTRL48 and XpSIS2-3, respectively), which may be caused by silencing of, or damage to, the *mCitrine* gene and/or a competitive disadvantage of mCitrine-positive cells to mCitrine-negative cells.

### Structural abnormalities induced by SCFs

To investigate X chromosome abnormalities following SCFs, we performed the dual-colored FISH on metaphase spreads. mCitrine-positive XpSIS2-3 sgF11 and XpCTRL48 sgF11 cells were sorted, and either harvested (D10) or re-cultured for another 7 d (D10–17) to follow the short- and long-term fate of the X chromosome, respectively (Fig 3A). Because mCitrine expression might cause chromosome abnormalities, we also analyzed XpSIS2-3 cells expressing retrovirus-delivered mCitrine as a control. The percentage of near-tetraploid cells and the ratio of centromere-positive X chromosomes to total chromosomes did not alter under all conditions (Fig S5A and B), suggesting that a single SCF does not cause numerical abnormalities.

Next, we observed several structural abnormalities of the X chromosome, including translocations (Fig S1I), chromosome fusions (Fig S5C), and extra-acentric X chromosomes. We categorized the extra-acentric X chromosomes into four sub-groups: fragment, scattering, ring, and extra-X translocation (Figs 3B–D and S5D). The fragment and scattering represent a single chunk (Fig 3B), and multiple fragments (Figs 3C and S5D) of acentric X chromosomes, respectively. Among these abnormalities, chromosome fusions (Fig S5C), as well as translocations (Fig S1I), were observed in untreated XpSIS2-3 cells and the level was similar across all conditions in XpSIS2-3 cells (Figs S5E and S6A and B), which makes it difficult to assess the effect of an SCF on these abnormalities. In stark contrast, we observed extra-acentric X chromosomes almost exclusively in XpSIS2-3 sgF11 D10 cells (Fig 3E). Whereas extra-X translocations are rare and not specific to XpSIS2-3 (Figs S5D and S6C), other extra-acentric X chromosomes are specific to XpSIS2-3 sgF11 D10 (Figs 3B–D and S6D–F), suggesting that these abnormalities are induced shortly after the generation of an SCF. The absence of these abnormalities after re-culturing suggests that either these abnormalities were repaired, or cells harboring these abnormalities were removed from the cycling population because of reduced fitness.

### Lineage analysis of mCitrine-positive cells by live-cell imaging

To further dissect cellular events following an SCF, we performed a lineage analysis by live-cell imaging of XpCTRL48 and XpSIS2-3 cells expressing either full-length mCitrine (mock control) or CRISPR/Cas9-sgF11 (Fig 4A). In the Cas9 and sgF11 expression condition, XpCTRL48 and XpSIS2-3 cells that became mCitrine-positive during live-cell imaging represent the first few rounds of the cell cycle after the *neoR* deletion (repair), and the generation of an SCF, respectively (Fig 4A and B, (1+x)th cycle). On the other hand, mCitrine-positive cells at the beginning of the imaging are in unknown rounds of the cell cycle after these events and categorized as (N+x)th cycle (Fig 4A and B). We analyzed the following characteristics of mCitrine-positive cells: interphase duration, mitotic duration, fading of the mCitrine fluorescence, and cellular abnormalities, which include cell death, MN formation, bi/multi-nuclei formation, tripolar mitosis, cytokinesis failure/furrow regression, and cell fusion (Fig 4C). We visualized individual lineages of mCitrine-positive cells as lineage trees with distinct symbols representing the cell cycle features (Figs 4C–F and S7–S12). For example, Fig 4D shows XpSIS2-3 sgF11 cells in the (1+x)th cycle. The corresponding lineage tree depicts interphase duration, mitotic duration, and cell division by a gray bar, a green bar, and a bifurcation, respectively (Fig 4D, right panel). Typical examples of fading of the mCitrine signal in the (1+x)th cycle (Fig 4E) and daughter cell fusion followed by tripolar mitosis and multi-nucleation in the (N+x)th cycle (Fig 4F) are also shown.

To assess the consequence of a single SCF, we calculated percentages of lineages that show the abnormalities for each condition (Figs 5A and S13A). Fading of the mCitrine was exclusively observed in XpSIS2-3 sgF11 cells, consistent with the notion that this fading is a consequence of an SCF (Fig S4A). We found that cell fusion and bi/multi-nucleation increased in XpSIS2-3 sgF11 in the (N+x)th cycle and that MN formation became prominent in XpSIS2-3 sgF11 lineages compared with other lineages (Fig 5A and B).

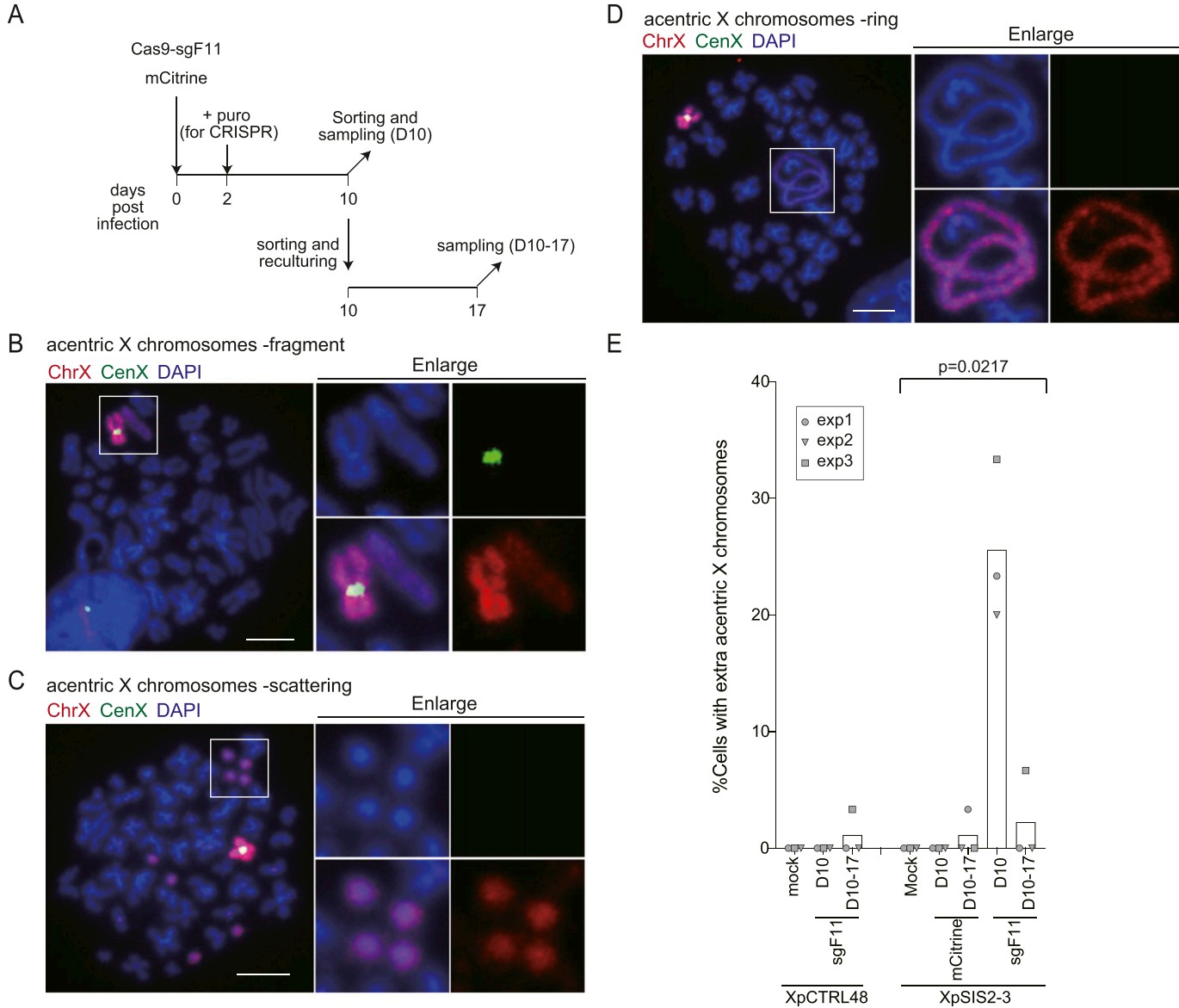

**Figure 3. A single SCF generates extra-acentric X chromosome fragments.**
**(A)** Schematic of FISH analysis. XpCTRL48 and XpSIS2-3 cells infected with lentivirus carrying CRISPR/Cas9 and sgF11 were selected with puromycin. XpSIS2-3 cells were independently infected with a retrovirus carrying the full-length *mCitrine* gene. mCitrine-positive cells were sorted at 10 d postinfection and harvested (D10), or re-cultured for 7 d and harvested (D10-17). **(B, C, D)** FISH images of fragment (B), scattering (C), and ring (D) phenotypes of abnormal extra-acentric X chromosomes in XpSIS2-3 sgF11 cells at 10 d postinfection. The images show the whole X chromosome (red), the X centromere (green), and DAPI (blue). Colors were adjusted on individual and merged images. Scale bar: 10 μm. **(E)** Percentage of cells carrying extra-acentric X chromosomes in indicated conditions. Bars represent mean from three independent experiments (n = 30 in each experiment). *P*-values were calculated by RM one-way ANOVA.
Source data are available for this figure.

To confirm the formation of MN involving the X chromosome, we performed the dual-colored FISH on interphase XpSIS2-3 sgF11 cells after sorting and re-culturing of mCitrine-positive cells. The FISH analysis revealed that XpSIS2-3 sgF11 cells show increased formation of MN that is positive for the whole X chromosome probe but negative for the X centromere probe (Fig 5C and D), whereas the same population of cells did not show an increase in the X chromosome-negative MN and the X centromere-positive MN (Fig 5C and D). We also found that the same population of cells

possesses a DNA bridge that stems from X chromosome signals between two interphase nuclei (Fig S13B), suggesting that the X chromosome bridge remains unresolved in the following G1 phase.

## Modeling of the fate of a single SCF

To statistically infer the impact of an SCF on MN formation in individual cells, we performed logistic regression analysis. We modeled the probability of MN formation ($q_n$) as a function of the linear predictor including

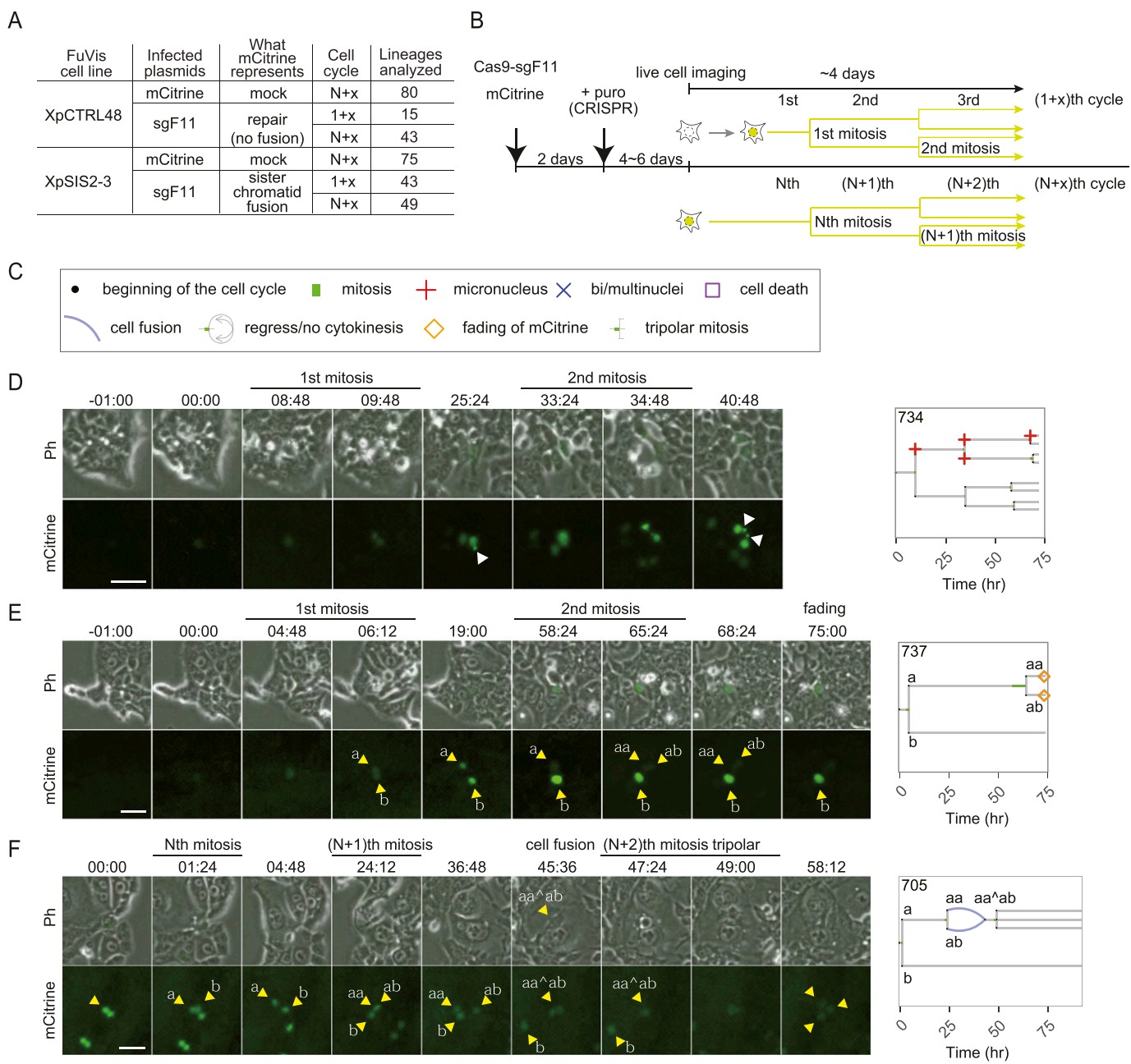

**Figure 4. Live-cell analysis of the fate of a single SCF.**
**(A)** A Summary of the live-cell imaging analysis. Indicated FuVis cell lines were infected with either retrovirus carrying pMX-*mCitrine* or lentivirus carrying *CRISPR/Cas9-sgF11*. **(B)** The cell cycle indicates the number of cell cycles after mCitrine expression, as shown in (B). **(B)** Schematic of the live-cell imaging analysis. N is an integer greater than 0, and x is an integer greater than or equal to 0. **(C)** Symbols representing cell cycle progression and cellular abnormalities in lineage trees. **(D, E, F)** Live-cell images of the fate of mCitrine-positive XpSIS2-3 sgF11 cells (left) and corresponding lineage trees (right): cell division during the (1+x)th cell cycle with MN formation (white arrowhead) (D), fading of mCitrine in one of two sister cell lineages (yellow arrowheads) during the (1+x)th cell cycle (E), and sister cell fusion followed by tripolar mitosis during the (N+x)th cell cycle (F). Yellow arrowheads with alphabetical labels (a, b, aa, ab, aa^ab) denote lineage orders, where aa^ab indicates a fused cell. White arrowheads denote MN. Scale bar: 50 μm.
Source data are available for this figure.

the following explanatory variables: SCF (XpSIS2-3 sgF11); RNF, repair/no fusion (XpCTRL48 sgF11); STG, cell cycle stages after mCitrine expression (1+x or N+x); and SIS2-3, cell line effect (XpSIS2-3 compared with XpCTRL48) (Fig 5E and Supplemental Data 4). To examine if each lineage of mCitrine-positive cells possesses distinct tendency to generate MN, we

also integrated the tendency as hierarchical structure into the parameters (lin; unknown individuality of each lineage) and constructed six models, which we applied to the data obtained from the movie analysis to estimate the posterior distribution of the parameters (Fig 5E and Supplemental Data 4). We calculated Widely Applicable Information

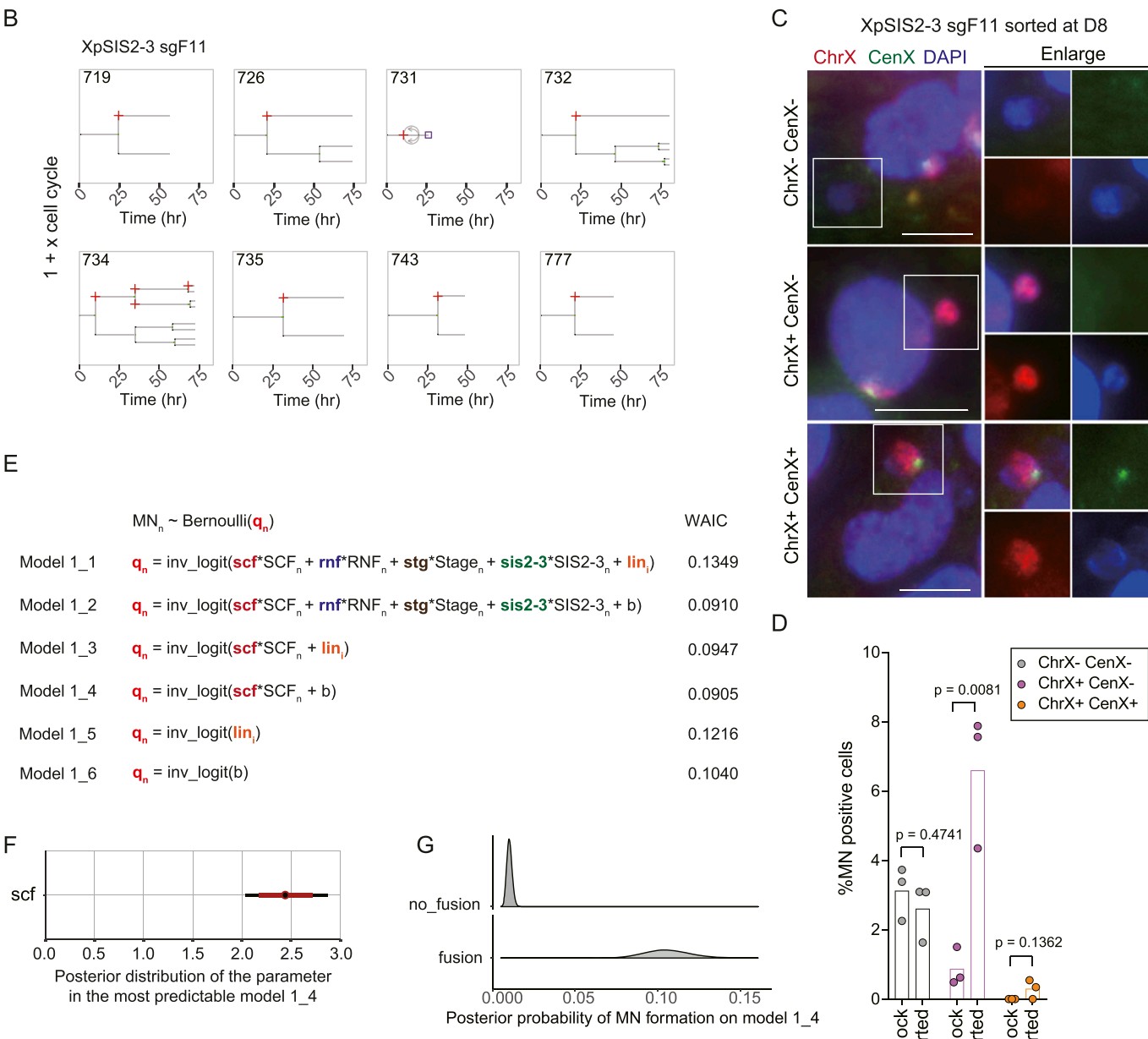

**A**

| | | | %Lineages that show indicated abnormalities | | | | | | | |
|---|---|---|---|---|---|---|---|---|---|---|
| | | | No mitosis | Mitotic Delay | Micronuc | Bi/multinuc | Cell death | Regression | Cell fusion | Fading |
| CTRL48 | mCit | N+x | 11 | 15 | 19 | 6 | 15 | 11 | 5 | 0 |
| | sgF11 | 1+x | 7 | 13 | 0 | 13 | 13 | 7 | 0 | 0 |
| | | N+x | 28 | 14 | 12 | 5 | 26 | 2 | 2 | 0 |
| SIS2-3 | mCit | N+x | 3 | 7 | 19 | 4 | 0 | 3 | 3 | 0 |
| | sgF11 | 1+x | 28 | 2 | 33 | 0 | 7 | 2 | 5 | 7 |
| | | N+x | 12 | 8 | 39 | 22 | 4 | 6 | 16 | 27 |

**B** XpSIS2-3 sgF11

**C** XpSIS2-3 sgF11 sorted at D8

**E**

$MN_n \sim Bernoulli(q_n)$ — WAIC

Model 1_1   $q_n = inv\_logit(scf*SCF_n + rnf*RNF_n + stg*Stage_n + sis2\text{-}3*SIS2\text{-}3_n + lin_l)$   0.1349

Model 1_2   $q_n = inv\_logit(scf*SCF_n + rnf*RNF_n + stg*Stage_n + sis2\text{-}3*SIS2\text{-}3_n + b)$   0.0910

Model 1_3   $q_n = inv\_logit(scf*SCF_n + lin_l)$   0.0947

Model 1_4   $q_n = inv\_logit(scf*SCF_n + b)$   0.0905

Model 1_5   $q_n = inv\_logit(lin_l)$   0.1216

Model 1_6   $q_n = inv\_logit(b)$   0.1040

**F** scf — Posterior distribution of the parameter in the most predictable model 1_4

**G** Posterior probability of MN formation on model 1_4

**D** %MN positive cells

**Figure 5. A single SCF leads to micronuclei formation.**
**(A)** A heat map representing percentages of lineages that possess the indicated abnormalities. No mitosis indicates a lineage that did not enter mitosis during the movie. Mitotic delay represents a lineage that engaged in at least one mitosis longer than 2 h. **(B)** Lineages that show MN formation during the (1+x)th cell cycle in XpSIS2-3 sgF11. Red cross, purple square, and curved arrows represent MN, death, and regression, respectively. The numbers denote lineage ID. **(C)** FISH images of MN in mCitrine-positive XpSIS2-3 sgF11 cells sorted at 8 d postinfection and re-cultured for 2 d. The images show the whole X chromosome (red), the X centromere (green), and DAPI (blue). Colors were adjusted on individual and merged images. Scale bar: 10 $\mu m$. **(D)** Percentage of cells carrying indicated MN. Bars represent mean from three independent experiments (n = 133, 161, 207 and 291, 292, 284 for mock and sorted, respectively). P-values were calculated by a two-tailed t test. **(E)** Six model structures

Criterion (WAIC), a statistical measure to estimate the generalization error of the models to the unknown distribution that generated data (Watanabe, 2010). The assessment by WAIC demonstrated that the most predictive model is model 1_4 (Fig 5E and Table S3), indicating that implementation of neither lineage individuality nor linear predictors other than SCF improved the predictability of the model. The estimated posterior distribution of the coefficient parameter *scf* in model 1_4 (median, 2.44, and 2.5 percentile, 2.03) indicates that SCF has a positive effect on MN formation (Fig 5F). Average posterior probabilities in the absence and the presence of SCF (0.0102 and 0.106, respectively) suggest that a single SCF increases the probability of MN by 10.3 times on average (Fig 5G). The inferred distribution of the parameters and average posterior predicted probabilities of MN formation in the second predictive model 1_2 indicate the positive effect of an SCF, and minor or no effects of repair/no fusion, cell cycle stage, and cell line on the probability of MN formation (Fig S13C and D). Thus, the predicted probabilities of MN in both models 1_2 and 1_4 indicate that MN formation depends on a sister chromatid formation in the FuVis system.

To address if MN formation has any negative effect in the descending lineage, the fates of matched sister–cell pairs, in which one of them displayed MN, were compared. We found that cells with MN possessed a higher probability of subsequent abnormalities (0.271), including fewer mitoses and increased incidence of MN, regression, cell fusion, cell death, and mitotic delay, than MN-negative sister lineages (Fig 6A). To further estimate the effect of MN formation on cell cycle duration, we assessed the log–normal distribution for models of interphase duration in terms of WAIC (Table S3). We set the log of the median (*mu*) of distribution as a function of the presence of MN and other variables (Fig S13E). Because both MN and interphase duration can be affected by the other variables (SCF, RNF, SIS2-3, and STG), we assumed that the co-efficient parameters (*scf*, *rnf*, *sis2-3*, and *stg*) are confounding factors, which can affect interphase duration indirectly through MN formation (Fig S13F). The best performing model among the six models in terms of WAIC was model 2_2, which included all confounding factors (Fig S13E). Therefore, our assumption about confounding factors to infer the causality between MN and interphase duration is considered to be appropriate in terms of WAIC (Supplemental Data 4). We also assessed the exponential and the γ distributions with the same parameters as model 2_2 and found that the log–normal model performed best among the three (Fig S13E and Table S3). The posterior distribution of the coefficient parameter of MN (micro) in model 2_2 (median, 0.820; 2.5 percentile, 0.675) (Fig S13G), indicates that the presence of MN prolongs interphase duration 2.27 (=$exp(0.88)$) times longer than its absence. The shapes of the predicted distribution of interphase duration, shown with 25, 50, and 75 percentile bars, suggest that the presence of MN delays and broadens the distribution of interphase duration (Fig 6B). Comparison of the interquartile range (IQR) and Kolmogorov-Smirnov distance of each distribution indicate that the presence of MN significantly increases the variability of interphase duration and that XpSIS2-3 sgF11N + x shows the highest IQR (Fig 6C). These results and analyses support the notion that cells with MN induced by an SCF hazard the strongest destabilization of the cell cycle.

## Discussion

It was previously challenging to assess the consequence of a single defined chromosome fusion, especially in the first few cell cycles after induction. Here, we have developed the FuVis-XpSIS system, a potent tool that allows visualization of a cell that possesses a single SCF at the short arm of the X chromosome (Fig 1). By using the FuVis-XpSIS, we have shown that a single SCF causes MN formation in the first few cell cycles and transiently generates extra-acentric X chromosomes (Figs 3 and 5).

### Construction and assessment of FuVis

We successfully isolated XpSIS2-3 and XpCTRL48 clones that harbor a single copy of the cassette at the distal end of the Xp arm (Figs S1A–H and S3A–F), although the frequency of multi-copy integrations was quite high. We do not know the mechanism of such multi-copy integrations; however, the sister cassette integration, which used a slightly longer telomere-proximal homology arm (43 bp longer than ctrl cassette), resulted in more frequent multi-copy integrations (Table S2), implying that using shorter homology arm may reduce the frequency of multi-copy integrations. The single-copy integration minimizes a possible false-positive mCitrine expression resulted from Cas9-induced rearrangements in the sister cassette. The generation of SCF upon the expression of Cas9-sgF11 in XpSIS2-3 was supported by multiple different observations including direct visualization of SCF by metaphase FISH (Figs 1D and S2B), anaphase bridge formation (Fig 1E), interphase bridge formation (Fig S13B), fusion junction sequences (Fig 1F), fading of mCitrine expression (Figs 2E, 4E, 5A, and S4) and the acentric X chromosome abnormalities (Figs 3, S5D, and S6D–F). We assume that the percentages of cells that acquired the expected rearrangement in XpSIS2-3 and XpCTRL48 are underestimated because extensive truncation at the fusion junction results in SCF and repair without mCitrine expression. Indeed, background expression of mCitrine in XpCTRL48 cells is diminished by the expression of Cas9-sgF11 in some cells (Fig 2C), suggesting a loss of mCitrine gene through the repair process in this population. Such potential underestimation, however, does not affect the interpretation of the results since we can focus our analysis on mCitrine-positive cells. The limitations of the current FuVis are that XpSIS2-3 carries background level of translocations at Xp subtelomere (Fig S1J) and that efficiency of mCitrine induction (i.e., detection of SCF) is relatively low (~1%). The translocations should not affect the SCF formation, whereas it makes it difficult to interpret the chromosome

constructed to explain a random variable micronuclei (MN) from the Bernoulli distribution (top). Small and large capitals on the right side indicate parameters and variables (dummy variables) obtained from data (0 or 1 in source data for Figs 4–6), respectively. Widely Applicable Information Criterion per sample values on the right. See the Materials and Methods section and Supplemental Data 4 for details. **(F)** The posterior distribution of the parameter *scf* inferred from the most predictable model (model 1_4) with median (black circle) and 50% (red bar) and 95% (black bar) credible intervals. **(G)** Distribution of posterior predicted probabilities of MN formation on the model 1_4.
Source data are available for this figure.

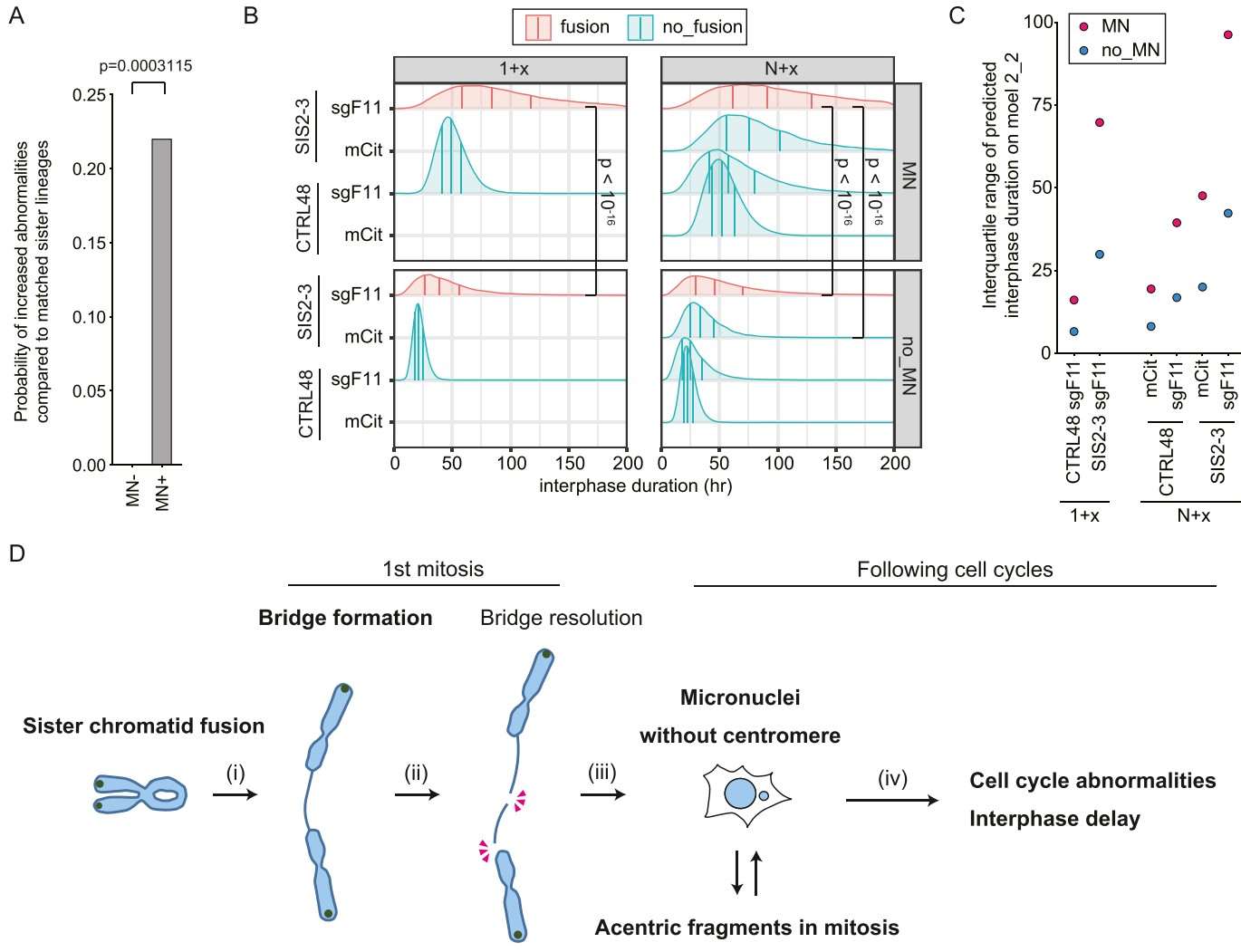

**Figure 6. Modeling of the effect of MN on interphase duration.**
**(A)** The inferred probability of increased abnormalities compared with the sister lineage. Cell cycle abnormalities were compared between 43 matched sister pairs, one of which possesses MN, selected from all lineage trees of mCitrine-positive XpSIS2-3 (167 trees) and XpCTRL48 cells (138 trees). MN+ descendants showed increased abnormalities in 12 lineages, whereas none of MN descendants did. *P*-value was calculated by the Chi-square test. **(B)** Predictive distribution of interphase duration with the model 2_2 shown in Fig S13E. The values of parameters and ς inferred from the most predictable model 2_2 were used to build the distribution of interphase duration in the indicated conditions. The three vertical lines in each distribution indicate 25, 50, and 75 percentiles from left to right. *P*-values were calculated by the Kolmogorov-Smirnov statistic. **(C)** The interquartile range (IQR) of predicted interphase duration with the most predictable model 2_2. **(B)** IQR was calculated by subtracting 25 percentile from 75 percentile in each condition in (B). **(D)** A model for the cellular fate of a single SCF. Please refer to the main text for the detailed explanation. All phenotypes other than bridge resolution were directly observed in FuVis system.
Source data are available for this figure.

abnormalities involving Xp subtelomere in XpSIS2-3 sgF11 cells. Improvement of these limitations is an important future issue.

### The implication of the statistical modeling

We applied generalized linear models to the live-cell imaging data to assess multiple different experimental variables in the statistical models. We also constructed the hierarchical Bayesian models to implement unknown lineage individuality in the statistical models. The lineage individuality assumes that, if a given cell possesses a particular abnormality, ascending and descending cells in the same lineage tend to show the same abnormality, which is a reasonable

assumption but often ignored in statistical analyses in other studies. This assumption needs to be addressed because it is important to avoid over- or underestimation of the effect of variables of interest when experimental data have a clustered structure (Galbraith et al, 2010; Lord et al, 2020). However, a comparison of WAIC demonstrated that the implementation of the lineage individuality failed to improve the predictability of the models (Table S3). This result suggests that individual lineages do not tend to possess similar abnormalities, although it is also possible that the live cell data do not contain enough number of cells in each lineage to assess lineage individuality in our model. We, therefore, focused on the most predictable model without

individuality in both models 1 and 2, which infer the effect of experimental variables in the sense of minimizing prediction error.

The most predictable models indicate that SCF, but not other experimental variables, increases the probability of MN formation by 10.3 times (Fig 5G) and that MN delays the average interphase duration by 2.3 times (Fig S13G). In model 2, we could also assess the effect of MN on the stability of the cell cycle. We used IQR of the interphase duration as a measure of cell cycle stability because when a cell incurs an abnormality, the cell is assumed to delay or even halt the cell cycle, which broadens the distribution of the interphase duration. The comparison of IQR of the predicted interphase duration on the most predictable model 2_2 suggests that MN formation destabilizes the cell cycle, especially in XpSIS2-3 sgF11 conditions (Fig 6C). Because the IQR of XpSIS2-3 sgF11 N + x condition is higher than that of XpSIS2-3 sgF11 1+x, we assume that the negative impact of MN on the cell cycle accumulates during cell cycle progression. Our results also suggest that an SCF negatively affects the cell cycle even in the absence of MN (Fig 6C). Therefore, our Bayesian statistical analyses allowed us to quantitatively infer the experimental effects and their uncertainty with the most predictable model (Figs 5F and S13G) and indicate that a single SCF, but neither the DNA damage repair process nor other experimental variables, destabilizes the cell cycle through cumulative effects of MN formation.

### The fate of a single SCF

The powerful FuVis-XpSIS system allows us to model the fate of a single SCF (Fig 6D). A single SCF causes an anaphase bridge (Fig 6D-i). Our results suggest that a single SCF is not sufficient to cause cytokinesis failure (Pampalona et al, 2012) nor mitotic delay (Hayashi et al, 2015) in HCT116 cells. We indeed detected a persisted chromatin bridge between interphase nuclei (Fig S13B), resolution of which can occur through either enzymatical digestion (Maciejowski et al, 2015) or mechanical breaking (Umbreit et al, 2020) (Fig 6D-ii). The resolution of the elongated bridge at more than two sites can generate acentric chromosome fragment(s) that is not incorporated into the main nucleus, resulting in MN without centromere (Fig 6D-iii). The formation of MN is consistent with a previous observation that about 23% of cells with H2B–GFP–visualized chromosome bridges during the anaphase proceed to generate MN in the following G1 phase (Rao et al, 2008). The X chromosome scattering phenotype is consistent with reports that chromosomes in MN are prone to DNA damage during the S phase due to abnormal nuclear envelope and subjected to fragmentation, which potentially results in chromothripsis (Crasta et al, 2012; Zhang et al, 2015; Ly et al, 2017). Such replication stress and DNA damage may contribute to cell cycle destabilization in MN-positive cells (Fig 6D-iv). Other subtypes of acentric chromosomes (i.e., fragment and ring) can be caused by the re-ligation of scattered chromosomes, abnormal amplification of acentric chromosome in MN, or both. We noticed that these subtypes of acentric X chromosomes are often more massive than the parental X chromosome, and possess reduced FISH signal compared with the parental X chromosome (Fig 3B and D). Such characteristics suggest that these acentric X chromosomes failed in mitotic condensation and support the notion that they derived from MN (He et al, 2019). Therefore, we propose that even a single SCF can generate MN and acentric chromosome fragments, which is followed by cell cycle destabilization. Sporadic SCF is indeed observed in tumor samples (Tanaka et al, 2014) and potentially leads to chromosome rearrangements and tumorigenesis.

### Perspective of FuVis

Fluorescent proteins have been widely applied to various studies, from protein labeling to biological sensors (Chudakov et al, 2010). We have expanded the applications of fluorescent proteins, namely a sensor of the specific chromosome rearrangement. The system described here can be applied to other rearrangements including specific translocations and different types of chromosome end-to-end fusion, such as inter-chromosome and intra-chromosome fusions, all of which may contribute to chromosome-driven cellular transformation, tumor development, and developmental disorders potentially through distinct mechanisms (Yip, 2015; Maciejowski & de Lange, 2017). Such expanded FuVis systems will provide unprecedented tools for direct visualization and short- and long-term trace of specific chromosome rearrangements in user-defined cellular contexts, including mouse models and recently developing organoid models.

# Materials and Methods

### Cell culture

Human colon carcinoma HCT116 cells (ATCC: American Type Culture Collection) and their derivatives were cultured in DMEM (Nissui Pharmaceutical) supplemented with 0.165% $NaHCO_3$, 2 mM L-glutamine, 1 mM penicillin/streptomycin, 2.5 $\mu$M plasmocin (InvivoGen), and 10% fetal bovine serum. All cells were grown at 37°C, with 5% $CO_2$ and ambient $O_2$.

### Plasmids

All plasmids used in this study are listed in Table S4. For cloning of sister cassette and control donor plasmids used for genomic integration, DNA fragments containing split mCitrine and CRISPR/Cas9 target sites were artificially synthesized (Eurofins Scientific), and used for the subsequent cloning. The potential CRISPR/Cas9 target sites were chosen from published non-targeting control sgRNA sequences that do not target the human genome (Wang et al, 2014). The neomycin resistance gene, tandem cHS4 insulators (West et al, 2004), and Xp subtelomere genomic sequence for homology templates were added during the cloning process. The resulting pMTH397 and pMTH729 constructs (Supplemental Data 2 and 3) were used for the generation of XpSIS and XpCTRL clones, respectively. For pMTH397, 1,022-bp telomere-distal and 723-bp telomere-proximal sequences were used for homology arms, whereas in pMTH729, the 1,022 telomere-distal and shorter 680-bp telomere-proximal sequences were used, which might underlie the difference of the frequency to obtain clones with multiple copies of the cassette in XpSIS and XpCTRL cells. Sequence information for other plasmids is available upon request.

## CRISPR/Cas9-mediated homology-directed DNA cassette integration into genomic DNA

For DNA cassette integration, we used HCT116 cells because they possess relatively stable near-diploid chromosomes (n = 45), display highly efficient HR, and carry only one X chromosome after having lost the Y chromosome. The CRISPR Design Tool on Feng Zhang lab's website (currently shut down) and the Cas-OFFinder program (Bae et al, 2014) (www.rgenome.net/cas-offinder/) were used to choose the target site for integration with minimal off-target sites. CRISPR/Cas9-based genome targeting was performed as described previously (Natsume et al, 2016). Briefly, HCT116 cells were plated in six-well plates 1 d before transfection. Using FuGENE HD reagent (Promega), the donor plasmids, pMTH397 or pMTH729, were transfected along with eSpCas9(1.1)-sgCHRXpYp-Subtel2 (pMTH393), which targets the chromosome Xp subtelomeric locus. 2 d post-transfection, the cells were collected, diluted, and plated on a 10-cm dish with 700 μM G418. Medium with G418 was refreshed every 3 d for 2 wk, and individual colonies were isolated. Genomic DNA from individual clones was obtained to assess genomic integration by conventional and qPCR (Applied Biosystems Veriti 96 Well Thermal Cycler, and Applied Biosystems StepOnePlus Real-Time PCR, respectively) using primers listed in Table S5, and Southern blotting as described below. For qPCR, genomic AAVS1 locus was cloned into a plasmid carrying sister cassette, which was used as a standard template to determine the relative copy number of sister cassette in XpSIS and XpCTRL clones. The products of conventional PCR were sequenced to confirm expected genomic integration.

## Southern blotting

Genomic DNA was digested with EcoRI, separated on a 0.7% SeaKem GTG agarose gel (Lonza), and transferred to an Amersham Hybond-N+ membrane (GE Healthcare Life Sciences). For the mCit-C probe, an 833-bp fragment was amplified by PCR using MTH384 and MTH417 as primers and pMTH393 as a template. The probe was generated by random labeling with $\alpha$-$^{32}$P dCTP and hybridized to the membrane at 63°C.

## Viral infection

The lentivirus particles were generated as described previously (Stewart et al, 2003) with minor modifications. Briefly, HEK293FT cells (Thermo Fisher Scientific) were transfected with transfer plasmids, psPAX2 (a gift from Didier Trono, #12259; Addgene) and pCMV-VSV-G (a gift from Robert Weinberg, #8454; Addgene), using polyethylenimine (PEI). The medium was replaced on the next day, and medium containing active lentivirus particles was collected on day 2 and day 3 posttransfection. For LentiCRISPR-sgEMPTY and LentiCRISPR-sgFUSIONs, cells were infected in growth media containing 8 μg/ml polybrene and lentivirus and cultured for 2 d. Puromycin was added to the culture at 1 μg/ml, and infected cells were selected for more than 2 d before analysis. The amount of lentivirus required for nearly 100% infection was determined empirically. All target sequences of CRISPR/Cas9 used in this study are listed in Table S1.

## FISH

Conventional FISH was performed as described previously (Cesare et al, 2015), with modifications as described below. For metaphase spread, cells were exposed to 100 ng/ml colcemid for 2 h and fixed in 3:1 Methanol/Acetic acid for 6 min. For interphase nuclei, cells were cultured on a coverslip coated with alcian blue, fixed in 3:1 Methanol/Acetic acid at −20°C for 10 min, and kept in the fixative at 4°C. The coverslip was air-dried overnight before hybridization. For X chromosome centromere and sister DNA cassette double-staining, a green fluorophore-labeled X centromere probe (XHO-10 X: Green; Chromosome Science Lab) and Cy3-labeled pMTH368 probe (on-demand probe; Chromosome Science Lab) were mixed in hybridization solution (Chromosome Science Lab) and used for hybridization at 70°C for 5 min according to the manufacturer's instructions. After overnight incubation at 37°C, the slides were washed in 2× SSC for 5 min at RT, 50% formamide/2× SSC for 20 min at 37°C, and 1× SSC for 15 min. The same hybridization and washing protocols were used for the X centromere and control DNA cassette double-staining with the XHO-10 X: Green probe and Cy3-labeled pMTH727 probe (on-demand probe; Chromosome Science Lab). For whole X chromosome and X centromere double-staining, orange fluorophore-conjugated X chromosome painting probe (XCP X orange; Metasystems) and green fluorophore-conjugated X chromosome centromere and orange fluorophore-conjugated chromosome Y centromere-specific probes (XCE X/Y; Metasystems) were used according to the manufacturer's instructions. Because male-derived HCT116 cells lost the Y chromosome, the XCE X/Y probe did not give any orange signal. In the structural abnormality analysis, an inter-chromosome fusion was distinguished from translocation by the presence of a narrow chromosomal region in a non-X chromosome, which suggests the presence of the centromere. Images were taken by a BZ-X710 all-in-one fluorescence microscope (KEYENCE) equipped with a 100× 1.45 NA oil CFI Plan Apo Lambda objective (Nikon). Blue, green, orange, and red fluorescence were detected with DAPI-optimized (ex: 360/40 nm, em: 460/50 nm, dichroic: 400LP), GFP-optimized (ex: 470/40 nm, em: 525/50 nm, dichroic: 495LP), TRITC-optimized (ex: 545/25, em: 605/70, dichroic: 565LP), and TexasRed-optimized (ex: 560/40 nm, em: 630/75 nm, dichroic: 585LP) filter cubes (M square), respectively. Individual color channels were adjusted for merged images.

## Flow cytometry

Cells were harvested by trypsinization, resuspended in 1× PBS with 0.1 mM EDTA, and filtered through a 5 ml polystyrene round-bottom tube with a cell-strainer cap (Corning) before passing through the FACSAria III flow cytometer/cell sorter (Becton Dickinson). Dead cells were excluded by positive PI-staining, and we gated single cells by their low FSC-W value before analysis and sorting. mCitrine-positive cells were detected by a 488 nm laser and 530/30 filter set.

## Fusion junction analysis

For SCF and repair junction analysis, the genomic DNA extracted from mCitrine-positive and the whole population of XpSIS2-3 sgF11 D10 cells was subjected to PCR using MTH672 and MTH673, and

MTH672 and MTH803 as primers, respectively. PCR products were gel-purified and cloned into an EcoRV site of plasmid pBSII by the In-Fusion cloning kit (Takara), followed by sequencing of individual clones. A deletion longer than ~750 bp on one side of the SCF junction abrogates splicing between the N-ter and C-ter of *mCitrine* and cannot be detected. For control cassette repair (*neoR* deletion) junction analysis, the whole population of XpCTRL48 sgF11 cells were harvested at 10 d postinfection and the genomic DNA was subjected to PCR with primer set MTH672 and MTH806. All primer sequences are listed in Table S5.

### Live-cell imaging

Live-cell imaging was performed in conventional cell culture dishes or plates placed on the BZ-X710 all-in-one fluorescence microscope (KEYENCE) equipped with a stage-top chamber and temperature controller with built-in $CO_2$ gas mixer (INUG2-KIW; Tokai hit), and a 10× 0.3 NA air CFI Plan Fluor DL objective (Nikon) at 37°C and 5% $CO_2$. mCitrine expression was detected with a metal-halide lamp and YFP-optimized filter cube (ex: 500/20 nm, em: 535/30 nm, dichroic: 515LP) (M square). Images were captured by the BZ-H3XT time-lapse module typically every 6–12 min for at least 66 h. The fate of mCitrine-positive cells was inspected manually. For all mCitrine-positive cells, the beginning time of mCitrine expression, the time of nuclear envelope breakdown and cytokinesis, and the time of abnormalities, including cell fusion, cell death, mitotic slippage, and tripolar mitosis were recorded. MN or multi-nuclei, including a binuclei phenotype, were determined by mCitrine localization. For XpSIS cells, the time when mCitrine became too faint to observe was also recorded. The fates of mCitrine-positive cells for all different lineages were recorded as a tidy data set (source data for Figs 4–6) and used for the statistical analysis and lineage tree visualization (Supplemental Data 1).

### Statistical analysis

We summarize the statistical framework here. Details are discussed in Supplemental Data 4. We adopted "high-level descriptive (top-down) statistical models" (Wilkinson, 2009) to the live-cell imaging data. In the analysis, each lineage has a group of cells, and it would be natural to assume that the cells share some common background characteristics affecting the observations such as the MN formation and interphase duration. If we ignore the cluster structure and consider each observation for each cell as a random variable subjected to independent and identical distribution, that would lead to a bias to the interpretation of the data (Galbraith et al, 2010; Lord et al, 2020). Therefore, we explicitly implemented the clustered or hierarchical structure into the statistical models. Furthermore, we constructed the alternative non-hierarchical model as well as other possible alternatives.

The probability of MN formation ($q_n$) is parametrized, and the MN is modeled as a random variable taken from Bernoulli distribution. The $q_n$ is linked by the *inv_logit*, inverse logit function, and following parameters: *scf*, the coefficient of SCF; *rnf*, the coefficient of repair (no fusion); *stg*, the coefficient of cell cycle stage after mCitrine expression (1+x or N+x); *sis2-3*, the coefficient of the cell line (SIS2-3 compared to CTRL48); $lin_i$, $i = 1,…, N_{lineages}$, intercepts assigned to

each lineage representing individuality (unknown cellular characteristics shared in each lineage, normal and student *t* distribution were assessed); and *b*, a bias parameter. Large capitals indicate variables (dummy variables) obtained from data (0 or 1 in source data for Figs 4–6). The $Int\_duration_n$ is modeled as a random variable taken from the log normal, the exponential, or the γ distributions and the coefficient of MN (*micro*) in addition to the parameters described above. The other parameter ϛ was assumed to differ among each experimental condition. We applied the models to the data and built predictive distributions on each model to make them approximate the unknown distribution that generated the data q(x), such as whether MN formed or not ($MN_n$, in Fig 5E).

The models and predictive distributions were defined and implemented by a probabilistic programming language Stan (Carpenter et al, 2017). We computed through the package "rstan" in the statistical computing environment R. In the programs, WAIC values were calculated to estimate the appropriateness of the predictive distributions p(x|Data) to q(x) (Watanabe, 2010). Thus, the smaller the WAIC of a model is, the closer the model is to the q(x). The assessment of the model is from the point of view of "prediction" originally proposed in the theory of Akaike information criterion (AIC) (Akaike, 1974), and WAIC is an extended version of AIC in a Bayesian framework (Watanabe, 2018; Harada et al, 2020). WAIC is applicable to the models that implement parameters whose posterior distribution does not resemble any normal distribution, which include the GLMMs. The difference between the WAIC values is meaningful, not the WAIC values themselves. If the difference in the values of WAIC between the two models is greater than one in the scale of AIC, the difference in the values of WAIC is considered to be significant (Sakamoto et al, 1986). Note that, in figures, we write the WAIC in the scale of the generalization loss, not the deviance scale conventionally used in AIC. For example, the smallest difference of WAIC values between model 1_2 and model 1_4 is 0.00045269 (nat/sample) (Supplemental Data 4). The sample size of the dataset is 4,424. Thus, the difference in the AIC scale is 4.00540112 (0.00045269 × 4424 × 2 = 4.00540112), which is greater than one and considered to be significant. The code is available in Supplemental Data 1. For cell cycle abnormality analysis in Fig 6A, the chi-square test was performed (source data for Fig 6A). For comparison of some pairs of predicted distributions of interphase duration, we used Kolmogorov-Smirnov test (Supplemental Data 4). For comparison of two groups, we used the two-tailed *t* test. For comparison of more than two groups in Figs 3, S5, and S6, we used the RM (repeated measure) one-way ANOVA.

### Lineage tree visualization

The tidy data sets were used to visualize lineage trees of all cell lineages analyzed in the live-cell imaging (source data for Figs 4–6). The time points of cell division are marked by bifurcation with green bars representing mitotic duration. When an individual cell showed a sign of MN or bi/multi-nuclei formation, the beginning of the cell cycle is marked by the respective symbols. When a given cell showed a sign of cell death or fading of mCitrine, the time point of the events is marked by the respective symbols. Blue lines represent cell fusion events, which can occur both in an inter- and intra-lineage manner. The code is available in Supplemental Data 1.

The trees with cell fusion events were manually modified with Adobe Illustrator CC 2019 in Figs S7–S12.

## Data Availability

Supplementary files are found at https://doi.org/10.6084/m9.figshare.7929266.

## Supplementary Information

## Acknowledgements

We thank Robert Weinberg, Didier Trono, Feng Zhang, and Itaru Imayoshi for sharing plasmids, Masato Kanemaki for sharing methods, Diana Romero Zamora and Hiroya Fukuda for experimental support with cloning, sequencing, or both, Mari Tsuboi for her assistance with cloning and data acquisition from live-cell imaging, Jan Karlseder and Hisao Masukata for critical reading of the manuscript, and Fuyuki Ishikawa laboratory and Anthony J Cesare laboratory members for helpful discussions. K Kagaya is supported by grant from Grant-in-Aid for Scientific Research on Innovative Areas (19H05330). MT Hayashi is supported by grants from the Senri Life Science Foundation, the Uehara Memorial Foundation, the Daiichi Sankyo Foundation of Life Science, the Nakajima Foundation, The Mochida Memorial Foundation Research Grant, Grant-in-Aid for Young Scientists (A) (16H06176), Grant-in-Aid for Scientific Research on Innovative Areas (16H01406 and 18H04712), Grant-in-Aid for Scientific Research (B) (20H03183), and the Kyoto University Hakubi project.

### Author Contributions

K Kagaya: data curation, formal analysis, funding acquisition, investigation, visualization, and writing—review and editing.
N Noma-Takayasu: formal analysis and investigation.
I Yamamoto: formal analysis.
S Tashiro: formal analysis.
F Ishikawa: conceptualization and writing—review and editing.
MT Hayashi: conceptualization, data curation, formal analysis, supervision, funding acquisition, validation, investigation, visualization, project administration, and writing—original draft, review, and editing.

### Conflict of Interest Statement

The authors declare that they have no conflict of interest.

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
