## [Reviewer comments · Life Science Alliance]

Life Science Alliance

Chromosome Instability Induced by a Single Defined Sister Chromatid Fusion

Makoto Hayashi, Katsushi Kagaya, Naoto Takayasu, Io Yamamoto, Sanki Tashiro, and Fuyuki Ishikawa

DOI: <https://doi.org/10.26508/lsa.202000911>

Corresponding author(s): Makoto Hayashi, Kyoto University

Review Timeline:	Submission Date:	2020-09-22
	Editorial Decision:	2020-10-05
	Revision Received:	2020-10-12
	Accepted:	2020-10-13

Scientific Editor: Shachi Bhatt

Transaction Report:

Please note that the manuscript was previously reviewed at another journal and the reports were taken into account in the decision-making process at Life Science Alliance.

Referee #2 Review

Report for Author:

"Chromosome instability induced by a single defined sister chromatid fusion", by Kagaya et al. This manuscript is a re-submission of a manuscript I reviewed in 2019. It tries to develop a reporter system that is capable of detecting sister fusions in human cells. I find this subject very important and interesting and a good reporter would clearly help the field out. Alas, I felt that that the project was poorly described and the data supporting it were underwhelming and thus, I suggested that while a revised manuscript might be acceptable, the current one was not. This was the message apparently conveyed to the authors and to their credit, they have resubmitted a manuscript that is, at least, easier to read. Unfortunately (because it is clear the authors have done a lot of work for this project) the data are still underwhelming/difficult to interpret and so once again I have to conclude that this manuscript does not come up to this journal's publication standards. I have tried to elaborate my thoughts below.

Specific Comments:

Minor:

The authors are to be commended on putting in the effort to make their manuscript more grammatically correct and accessible.

Major:

line 104. As I stated earlier, on paper, this is a very elegant reporter system. Alas, the biology appears to not be cooperating with the authors. This is truly unfortunate.

line 123. "...an AAVS1 sequence on chromosome 19 was cloned into...". I do not understand this control. I can see trying to do qPCR with the endogenous allele (which should be diploid), but why are they doing qPCR with a cloned sequence? Did they do some sort of a standardized dilution curve with this that they didn't describe?

Line 126. Yes, clone #3 appears to be the only single-copy (i.e., usable) clone. This very low frequency is not encouraging for future attempts to introduce the reporter into an investigator's favorite cell line.

Line 131. "...about 30% of the XpSIS2-3 cells show translocation...". This is awful. I agree that the fact that the reporter is imbedded into the middle of another chromosome will not alter necessarily the authors' selection scheme, but it WILL alter all the downstream parameters they measure and try to interpret as being biologically significant. 30% is too high a bkg to obtain meaningful data. Can the authors not subclone the cells to find a subclone where this translocation is not evident??? Or does this just happen at high frequency even in a clonal cell line?

Line 143. This is clear data that a sister fusion will lead to a bridge. Nice.

Line 149. "...Alignment of cloned sequences to expected fusions and repair junctions...". What exactly are they aligning it to? How can there be "expected" fusion and repair junctions? The authors should state exactly how much deletion is actually possible before the selection cassette is impaired.

Line 151. Fig. 1E. I'm utterly perplexed by this figure. What is a "long" deletion and what is a "short" deletion. Are the authors trying to show that one side of junction has suffered longer deletions than the other? I simply have no idea what the authors did here.

Line 154. "...are processed by microhomology-mediated end joining...". First the deletions data in Fig. 1E are uninterpretable. Second, at least 20% of the junctions have insertions in them, which is decidedly NOT a feature of MMEJ. Though it is probably outside the scope of this paper (and a tad unfair since I didn't ask for it in my first review), but a genetic experiment (say doing this in a POLQ-null cell line) would be a much stronger piece of data to infer that this is MMEJ.

Line 170. "...possessed a duplication...". Sigh, the authors have 2 important cell lines and both have technical problems with them. Again, I agree that having a downstream duplication does not affect the Citrine selection, but it could easily affect the frequency of chromosomal abnormalities.

Line 179. "...repaired products with deletion (Fig EV2H)." These are only the large deletions; there will decidedly be lots of junctional diversity in the bands that look ~parental and these should be analyzed as well, or?

Line 183. "Fig EV2J. Again, I have no idea what is being graphed here - I think it is the deletions at each end of the junction, but I really can't tell.

Line 184. "...rely on MMEJ...". Here, ~35% of the events have insertions, which is not a feature of MMEJ. I do not follow the authors interpretation here at all.

Line 248. The authors have provided a lot of data in Figs. 4 & 5, but I don't know what I'm supposed to do with it/what it actually means.

Line 252. "...fading is a consequence of sister chromatid fusion (Fig. EV3A). I get that if the fusion breaks one chromosome with the marker will go to one cell and another cell will get the chromosome without the marker, which I think is what the authors mean by "fading". However, the cell with the marker should never fade, or? I was confused here.

Line 257. Late in the paper the authors introduce a 3rd cell line and again this one has duplications, obscuring its interpretation.

Line 271. These studies are then expanded with a largely computational/theoretical analysis of the live cell imaging. I appreciate that this is how the authors want to adjudicate the data in figures 4 & 5, but I'm poorly placed to judge if this is really a contribution or even if it is accurately done.

At the end, the authors seem to conclude that a sister fusion can give rise to micronuclei. While direct demonstrations are always nice, I don't think this is a surprise to anyone in the field and I do not think it is a significant enough contribution to warrant publication.

Referee #3 Review

Report for Author:

-General summary and opinion about the principal significance of the study, its questions and findings

This work aims to link chromosome fusions, a classical chromosome abnormality that is frequently observed in cancer, with other chromosomal abnormalities in order to establish causality. To do this, the authors have conceived and constructed an ingenious reporter system that develops nuclear fluorescence in cells in which the desired fusion has occurred, permitting them to follow the immediate fate of these abnormalities. This is a very valuable tool since it allows single fusions between defined loci at the time of the experimenters choosing. The authors present evidence that their FuVis tool can induce fusions between sister X-chromosomes of HCT116 cells and track the fusion lineage to monitor a variety of abnormalities, including micronucleus formation. After demonstrating that micronuclei are abnormalities observed specifically after induction of fusion, they use modelling to suggest that the fusion chromosome is the most likely cause of micronuclei and that micronuclei are the likely cause of cell cycle delays in the cells that carry them.

I thank the authors for the rewriting of the manuscript, the current version reads much easier. Since the first submission of the manuscript, two things have happened. Firstly, prompted by reviewer #2 request, the authors discovered that their clones had multiple integrations of the fusion reporter cassette. This was the likely the cause of some of the reported chromosomal aberrations, including the numerical aberrations. The revised manuscript has undergone substantial rework and now shows only an association of micronuclei with chromosome fusions, leaving it with a much clearer

message which is also reflected in the rework of the summary figure. The second is a recent detailed study from the Pellman group following the fate of chromosome bridges and the emergence of micronuclei from them in the subsequent cell cycle (Umbreit 2020). This paper overlaps with the current one. However, the FuVis system remains interesting to me as current chromosome fusion systems do not have a readout that informs about the successful fusion event in real time.

-Specific major concerns essential to be addressed to support the conclusions

My biggest issue is the inclusion of the SIS1-15 cell line which is known to have multiple FuVis integrations as well as the use of SIS2-3 which carries chromosome abnormalities to begin with. I agree that it is possible and maybe even likely that these do not affect the main conclusions but it would be great to confirm this in a cell line that has no chromosome aberrations to start out with or single FuVis integration. Or otherwise the authors need to make clear in the text all possible limitations the above could have.

-Could the authors please contextualize the use of SIS1-15, especially after the authors went through the process of identifying and disregarding all multi-integration cell lines earlier.

-Lines 118-119: Could the authors please discuss what is the off-target integration rate? How many of the 72 lines are positive for the presence of the mCitrine sequence but negative for the right integration junction? If all 72 lines do indeed have constructs integrated, why is the off-target rate so high?

-Lines 134-136 claim that the translocation of additional undefined chromosomal material to the FuVis construct location in 30% of X-Chromosomes will not affect its function. Do the authors know whether the construct located at the translocation site is still functional? Would cutting now not liberate a large acentric fragment in 30% of cells? If this is the case and this is the only copy of these undefined sequences, cells in this lineage may have defects such as mitotic delays that stem from the initial cut and loss of essential sequences, not from the subsequent chromosome fusion and micronucleus formation. In addition, while the cut would still lead to X-X fusion, the "liberated" fragment may now form micronuclei on top of those formed downstream of the X-X fusion leading to overestimation of total micronuclei. Could the authors please clarify this?

-Lines 139-143: I would like to know the frequency of the X-X fusion formation as well as the DNA bridges compared to the mock transfected line. The images show the fusion but given that there are X-X chromosome fusions in SIS2-3 in absence of fusion induction (EV4D "mock"), I would like to know how efficiently the system works. Does the frequency of new fusions correspond to the frequency of mCitrine positive cells? On the same line, why is there no increase in X-X fusions after induction in the chromosome analysis in EV4D? I presume they may have been lost by D10 but it would be good to know that it's not a result of inefficient induction of fusion.

-EV2B: Could the authors please explain why are all integrated CTRL constructs of different length?

-Also, could the authors please provide a thorough description of the construct to better understand the origin of the frequent multi-copy integrations? This would help so that other experimenters can make an informed choice about whether this would be the right tool for them.

-Minor concerns that should be addressed

Lines 193-201: I agree that, on activation of the fusion construct, a single mCitrine expressing gene is formed that can only be propagated to one of two daughters. However, there could be other reasons for the decrease in fluorescent cells in the mCitrine sorted cell population. For instance, the loss of mCitrine positive cells could be the result of silencing of or damage to the gene or a competitive disadvantage compared to the remaining ~10% mCitrine negative cells in the sorted population. Could this be commented in the paper?

Lines 258-260: If there are two adjacent integrations of the construct, this can be tested and

known with certainty. Is it possible to amplify the region and Sanger sequence it to test for tandem integrations and their orientation?

Lines 392-397: could this please be rephrased?

Fig4F: What I think should be a symbol for tripolar mitosis isn't discernable

Fig5A: I don't like the vastly different colors used for the bins. It creates the impression that 19% and 21% are two very different outcomes while 10% and 19% are the same outcome. If a color aid is to be used I would prefer a gradient of the same color where the intensity corresponds to the value. I also can't find a measure for experiment to experiment variation. The 0% micronuclei in the guide transfected CTRL48 control line and the 19% micronuclei in the mock transfected CTRL48 line could suggest considerable variability between these two negative controls.

Fig5B: Please add a legend for the symbols used in these lineage graphs

Referee #4 Review

Report for Author:

In this revised version authors have expanded their statistical modelling. I think the inclusion of the Kolmogorov-Smirnov tests for the interphase duration distribution now clearly show that the fusion event has a significant effect. With the new analysis, the two key inferences: (i) that fusion leads to more micronuclei and (ii) that fusion leads to longer interphase duration, are now more clearly demonstrated. This definitely constitutes an improvement of the paper, as it puts the conclusions on a more solid basis. As I said in the previous round of review, I think the problem and questions are interesting and challenging, so certainly worth pursuing. I am therefore in favour of publication for this study.

I am still thinking that the modelling section is not reader-friendly as rather than formulas essentially code excerpts are given in the Supplementary Material (and also in part of the figures). This hampers a bit readability at least in my opinion. It would be good - as an optional modification - to give the mathematical formulas for the distribution used somewhere, but I would no longer insist on this as necessary prior to publication at this point. The nomenclature used is sufficiently clear that the calculations can be reproduced by users familiar with R and this is probably sufficient.

As a final suggestion (still optional, but highly recommended), it would be good to discuss a bit more the confidence range of the WAIC values. As model 1_4, for instance, is selected in Fig. 5, how confident are we that its WAIC is really lower than the others. More explicitly, the WAIC value for model 1_4 is 0.0910, that of model 1_2 is 0.0905: is this truly meaning that model 1_4 is better or within the errors they are equivalent? The situation is similar to that of chi squared in linear regression: it is good practice to give an error on chi squared via e.g. bootstrapping, and I am thinking that it would be good to do the same here, or at least add a discussion about the expected error/uncertainty on the WAIC.

Referee #1 Review

Report for Author:

In this work, the authors develop a novel system to interrogate the effects of sister-chromatid fusion. The authors inserted a transgene with an inverted and split fluorophore onto the

hemizygous X chromosome in HCT116 cells, and then they trigger sister chromatid fusion by cutting within the transgene. The expression of the fluorophore can then be used to identify cells in which a fusion has occurred. The authors then report that these fusion events cause numerous intra- and inter-chromosomal rearrangements, and can also drive micronucleus formation.

The system is ingenious and I recommend publication for this manuscript, with two concerns. First, I believe that the manuscript would benefit from careful proof-reading for spelling and grammar. Secondly, can the authors estimate what fraction of mCitrine+ cells result from sister-chromatid fusion? Is it 100%? Or, are some mCitrine+ cells the result of Cas9-induced rearrangements in the transgene without sister-chromatid fusion? What do the authors observe if they trigger Cas9 cutting in cells arrested in G1, for instance?

Referee #2 Review

Report for Author:

"Chromosome instability induced by a single defined sister chromatid fusion", by Kagaya et al.

This manuscript investigates the formation of sister chromatid fusions and their resolution in human cells. The authors first establish a cute reporter system with which they should be able to detect sister chromatid fusion events. They demonstrate that they can detect such events and quantitate these at the molecular level. They then extend these studies by following individual fusion/repair events with time-lapse, single cell microscopy. These studies are then expanded with a largely computational/theoretical analysis of the live cell imaging. The major claim of the manuscript is that this is direct evidence that a single sister chromatid fusion event can generate genome instability.

I'm conflicted by this manuscript. On the plus side I find the topic of sister chromatid fusion to be interesting and important. In addition, I thought the reporter that the authors developed was cute and potentially powerful. On the downside, I thought that the presentation was confusing and that much of the experimental data which was presented was either weak or confusing. In addition, I was really turned off by the large computational section, but perhaps that is because that is outside my area of expertise. At the end, I liked the topic and I liked the approach (theoretically), however, so much of the actual data was underwhelming that I'm left not being very supportive of the paper. I could certainly envision where I might be supportive of a revised manuscript, but that would have to entail serious revisions. I have tried to elaborate my thoughts below.

Specific Comments:

Minor: With absolutely no ill-will intended, it is clear that English is not the authors' first language. The manuscript is littered with typographical errors and poor sentence structure and this detracted from the presentation. Indeed, some (much?) of my lack of enthusiasm for the manuscript could simply be because the authors weren't clear about what they did or how they did it. A revised manuscript would really benefit from serious input from someone whose mother tongue is English and/or significant editorial help.

Major:

line 58. "which are not equivalent". Actually, this is hard to adjudicate, or? It is possible that tumor samples also start off with multiple fusions, but only those cancer cells with a smaller number are fit enough to survive and thus what gets selected out in the tumor, is not representative of how the multitude of events that started tumor initiation.

Line 83. 2 Clones. Many more are mentioned in the Results and only one of these clones looks even remotely usable. This is thus not a very compelling demonstration of the utility of this system.

Line 90/91. Mechanistically it was never clear to me how the authors thought the non-X chromosome aberrations were occurring.

Line 99. "The development of MN depends exclusively on the sister chromatid fusion". I do not believe this interpretation. Historically, even lagging chromosomes, completely unrelated to sister chromatid fusions, can cause MN. This may have just been poorly expressed on the authors' part.

Line 112. Fig. 1a. On paper, this is a cute reporter with a lot of nice, well-thought out features.

Line 124. My confusion starts right away. The authors state they did 2 independent transfections to isolate at least 11 G418+ clones, but then in Sup. Fig. 1 it looks like only 4 of them are actually what they want. If this is true, they need to state this! I'm guessing they then took 2 of them (clones 15 and 36) and then performed the CRISPR/Cas9 expression to try and induce the sister chromatid fusions. If this is true, they need to state this. What they wrote was, "We performed a second screening...". This is just confusing/doesn't make sense. It makes it sound as if they were trying to isolate more G418+ clones, but I think what they meant is that they were looking for G418>mCitrine conversion. Because the initial portion of this MS is so poorly presented, I became confused as to what was what and lost much of my enthusiasm for the paper.

Line 135. Clone 36 clearly looks as if it has multiple copies of the array integrated at Xp (Sup. Fig. 1f). If this is true, why would the authors ever use this clone? Their assay, to have any meaningfulness has to be a single copy integrant. For example, if two copies of the cassette integrated in a head to tail fashion the cells could become mCitrine-positive without any sister chromatid cohesion. To my eye, it looks like only clone 15 may be useful.

Line 135. FISH (Fig. 1c and Sup. Fig. 2a). Here (and disappointingly elsewhere) it seemed that the authors were glossing over experimental problems. Thus, while I certainly see a FISH signal at the end of the chromosome, I'm curious as to why the fluorescent intensity is much stronger than that of its sister? They should be precisely equivalent, or? Moreover, it is clear that the FISH signal on the stronger-hybridizing chromosome is split. What is the explanation for this? Is this unequal T-SCE? Was there a duplication of some part of the cassette? The vagueness of the molecular structure of these clones strongly detracts from their utility.

Line 141. Fig. 1d. I don't see what the authors want me to see. The CEN spots and the DAPI are OK. Why is there only 1 telomeric signal??? Do the other 3 ends not have telomeres? And the Sis-cassette staining looks like it is coming from the middle of the chromosome, or? There is nothing that is convincing about this data. Similarly, in Sup. Fig. 2a/b, what am I suppose to be looking at? What are there multiple signals of the cassette in the control cell line? None of this made much sense to me.

Line 150. What are the slower migrating bands in Sup. Fig. 2f?

Line 154. Somewhere the authors need to define their definition of microhomology. The most current definition is a microhomology of 3 nt or more. Is this the definition they were using?

Line 163. I, again, like the reporter and thought it was imaginative and cute. However, it is not really formally comparable to the experimental vector since it involves the formation of 2 DSBs and the loss of intervening sequences.

Line 174. Well the duplication of the HA is clearly not good, but it is probably acceptable. Once again, however, I have concerns about copy number. They need to be looking at single copy clones and they haven't proven to me that they have them.

Line 207. I like this control. It is believable.

Line 224. The authors here (and elsewhere) assume that the sister chromatid fusion that they have screened for is the only one in the cell, but there is really (devil's advocate) no way to prove that. Admittedly, I think the frequency of 2 or more SC fusions in the same cell is indeed likely remote.

Line 236. Why is de novo telomere addition undetectable in this analysis - because of the presumed lack of mCitrine expression?

Line 264. Can the authors be explicit about what they believe the mechanistic reason for "fading" is?

Line 276. This long section did nothing for me, but I readily admit that I'm completely unqualified to comment on either its veracity or its usefulness. Hopefully one or more of the other reviewers will have more expertise in this area.

Line 329. Again, I'm not convinced the authors have demonstrated what they contend they have demonstrated. If I believed them/the data more, I'd certainly be more supportive of this work.

Referee #3 Review

Report for Author:

I would like to preface this review by saying that I am not qualified to evaluate the statistical modelling and analysis in figure 5 and 6 as to whether they are applicable, correct and interpreted within reason.

Beyond this however I found the paper very thorough and significant.

As the authors correctly state in the introduction, while we often associate micronuclei with chromosome bridges, we have been lacking a system to introduce a defined chromosome fusion and track the immediate outcome. Previous methods often rely on inactivation of proteins which firstly can have pleiotropic effects and second triggers multiple, unpredictable and untimed fusion events. By generating a Cas9 based single-chromosome fusion system that is linked to a timely reporter of the desired event, the authors circumvent the limitations of previous experimental systems. In my opinion, this represents a major step forward in understanding the immediate outcome of a chromosome bridge and in deciphering causality between the many manifestations of genome instability. Such a system should be adaptable for different chromosomal locations and

potentially multiple defined fusion events and it will be interesting to see further developments. In the present manuscript the authors have made a number of important observations, which in my opinion are:

- Link between sister chromatid fusions and micronucleus formation.
- The lack of response of the cytokinesis checkpoint to a single chromosome bridge.
- The emergence of numerical chromosome abnormalities in addition to structural abnormalities from such a fusion event.

My suggestions are mostly related to form and some statements made.

Suggestions (Major):

-The introduction would benefit from some rewriting. I agree with the points made but it could read a little easier to read.

-The model in Figure 7 contains 10 references and 6 question marks. For a research article with such a clear message, the model figure seems unnecessarily complicated. I would suggest to reduce this for the sake of clarity and place this extensive summary in a future review.

Suggestions (Minor):

-The introduction states that micronucleus formation depends "exclusively" on the sister chromatid fusion (l.99). The presence of micronuclei in the control cells suggests that there are other sources. "Exclusively" should be toned down to reflect that it is exclusive among model parameters considered which do not include, for example, errors in chromosome-microtubule interactions.

-In figure 3D, the difference between "translocation" and "fusion with non-X" is not clear to me. Both images appear to show fusion chromosomes that contain X and non-X material with a single centromere.

-The lineage graphs seem to, in some cases, suggest fusion of two unrelated cells (e.g. 5B bottom center right or Supplemental 6, two down two right). Is spontaneous cell fusion a known occurrence in HCT116 cells? Since the authors are looking at a nuclear signal, is it possible that these represent regression events in which late cytokinesis fails rather than fusion events between unrelated cells? Is it possible that the cell fusion and regress events are not well discerned in this system?

-The labeling of the lineage analysis consists of 9 arbitrarily chosen symbols. It would be helpful for the reader if a legend of these symbols were given on every page that contains lineage graphs. Backtracking to the legend in figure 4 from supplemental 6-14 can become rather tedious.

-The wording in the discussion seems to imply that bridges resolve in G1 (l.402-403) although no cytokinesis delay is observed. In other words cytokinesis proceeds in a timely fashion, before G1, yet bridges persist into G1. I would suggest rephrasing this into bridges being resolved before G1, followed by timely cytokinesis and bridge healing in G1 (if this fits with the observations of the authors). I would also suggest to include, in addition to potential enzymatic cutting of the bridge (l.412), a possible mechanical breaking of the bridge.

Referee #4 Review

Report for Author:

This is a potentially interesting paper on the effect of stimulated sister chromosome fusion on chromosome instabilities and chromothripsis. As far as I know it is very difficult to stimulate

chromothripsis in a controlled fashion and therefore I think the current manuscript is certainly potentially interesting.

I am limiting my comments to the modelling section which is closest to my technical expertise. The authors use a statistical analysis based on Markov chain Monte Carlo to determine whether fusion and other variables have effect on cellular phenotype. They focus on the probability of formation of micronuclei (model series 1) and on the duration of interphase (model series 2).

Currently, the modelling section is not very easy to follow as it is spread over text, supplementary files and figure captions.

While Supplemental File 3 is clearer, it entails text interspersed with code and would not be an easy read for non-R users. It would therefore be good to make the text in the Supplementary Material more self-contained. To do so, it would be good to include there more explicitly all formulas used (e.g. inverse logit, Bernoulli distribution, log-normal distribution). This is only a presentation point, but it would render the reading easier for people wanting to understand the statistical analysis without running the R codes.

Regarding the micronuclei formation, it is unclear to me why this property was singled out for the analysis and why its observation was limited to a yes/no answer. In principle, I can imagine it should be possible to measure the size of the nucleus so as to give a continuum value which it would be in principle easier to correlate with fusion. For instance one could compute, I would say, the Kolmogorov-Smirnov distance between distributions of sizes with and without fusion (or perhaps with different combinations of variables). This might be naive but I'd like the authors to further motivate the choice of their observables and why they wanted it to be discrete (which also requires observer decision to call yes/no for a nucleus). The procedure used by the authors render it pretty difficult, at least for me, to determine the statistical significance of the inferences they draw (for instance in one of the cell types, CTRL48, if I understand correctly MN are not seen more often with fusion, hence statistical significance is important to me).

In the same section, I was unclear of how exactly the effect of the lineages were implemented. This is clearer from Supplemental File 3 but was not discussed in the text. In particular in Fig. 5 a mention of the random vector $b_{[lin]}$ should be made. As far as I understand the authors add a random bias vector depending on lineage index. It seems to me that the assumption that the bias was taken from a Gaussian might need more motivation. In principle I'd imagine one could have a different linear bias coefficient to be fitted by MCMC for each lineage, without assuming this to be a random variable taken from the same Gaussian distribution. In general, it is unclear to me why the authors bother including a lineage variable when this is treated essentially as a random perturbation. In Fig. 6a it would be needed to give a p-value or another assessment of the statistical significance of the difference between the abnormalities with/without MN.

When modelling interphase duration, can the authors say more about the distributions of times? Why should it be log-normal in the first place, in both the wild-type and with-fusion case? Can the authors compute the difference between distributions (e.g., via Kolmogorov-Smirnov) and say how likely any two distributions in Fig. 6b are to be different in a significant way? The best model here, 2_2, contains all variables so while it is predictive from the information theory point of view I think it leads to a limited insight with respect to, for instance, the MN analysis, where a simple model was selected which allowed a simple inference to be made. Maybe I am missing something here but if true this should be admitted more explicitly.

I think these concerns with the statistical analysis/modelling and statistical significance should be dealt with by the authors as the modelling is pretty crucial for their conclusions in this work.

October 5, 2020

RE: Life Science Alliance Manuscript #LSA-2020-00911-T

Dr. Makoto T Hayashi
Kyoto University
The Graduate School of Medicine
Yoshida-Konoe-cho
Sakyo-ku
Kyoto, Kyoto 60608501
Japan

Dear Dr. Hayashi,

Thank you for submitting your revised manuscript entitled "Chromosome Instability Induced by a Single Defined Sister Chromatid Fusion" to Life Science Alliance (LSA). We would be happy to publish your paper in Life Science Alliance pending final revisions necessary to meet our formatting guidelines.

For a brief overview, the manuscript was transferred from a partner journal, where it was reviewed twice, but the reviewers could not come to a consensus. With the help of the editors, the authors then transferred the revised manuscript, along with the referee's comments to LSA. LSA editors deemed the advance to be strong enough to consider the manuscript for publication, and sought opinion of one of the previous reviewers (see their comments below), who agreed that the manuscript is ready for publication.

Along with the points listed below, please also attend to the following,

- please make sure the author names entered in our system match the author names in your manuscript text
- please add a Summary Blurb/Alternate Abstract in our system
- please provide your manuscript text in editable doc format
- please upload your main and supplementary figures as singular files
- please rename your Expanded View Figures as Supplementary Figures (e.g. Figure EV1 = Figure S1) and adjust the figure callouts in the main manuscript text (Page 8: Fig EV1A and B = Fig S1A and B)
- please also rename your EV Tables to Supplementary Tables
- please add legends for your supplementary tables

To avoid unnecessary delays in the acceptance and publication of your paper, please read the

following information carefully.

A. FINAL FILES:

B. MANUSCRIPT ORGANIZATION AND FORMATTING:

Sincerely,

Shachi Bhatt, Ph.D.
Executive Editor
Life Science Alliance
<https://www.life-science-alliance.org/>
Tweet @SciBhatt @LSAJournal

Reviewer #3 (Comments to the Authors (Required)):

The manuscript has undergone a long series of revisions. I am now satisfied with the authors responses and changes to the text to respond to the concerns I have raised in my last revision to (previous journal). I am particularly pleased with the removal of data from potentially aberrant lines.

October 13, 2020

RE: Life Science Alliance Manuscript #LSA-2020-00911-TR

Dr. Makoto T Hayashi
Kyoto University
The Graduate School of Medicine
Yoshida-Konoe-cho
Sakyo-ku
Kyoto, Kyoto 60608501
Japan

Dear Dr. Hayashi,

Thank you for submitting your Research Article entitled "Chromosome Instability Induced by a Single Defined Sister Chromatid Fusion". It is a pleasure to let you know that your manuscript is now accepted for publication in Life Science Alliance. Congratulations on this interesting work.

Your manuscript will now progress through copyediting and proofing. ++ One edit to be made at the proofs stage - the manuscript is still missing a callout for Figure S2D, this needs to be added at the proofs stage ++

It is journal policy that authors provide original data upon request.

DISTRIBUTION OF MATERIALS:

Again, congratulations on a very nice paper. I hope you found the review process to be constructive and are pleased with how the manuscript was handled editorially. We look forward to future exciting submissions from your lab.

Sincerely,

Shachi Bhatt, Ph.D.

Executive Editor

Life Science Alliance

<https://www.lsjournal.org/>
